# Rapidly making biodegradable and recyclable paper plastic based on microwave radiation driven dynamic carbamate chemistry

Xinxin Yang [1,2,3,7], Le Yu [2,7], Bowen Zhang[3], Yongheng Wang[4], Xiangzheng Jia [4], Erlantz Lizundia [5], Chang Chen [6], Fuhao Dong[3], Luhe Qi[2], Lu Chen[2], Enlai Gao [4] ✉, Xu Xu[1] ✉, He Liu [3] ✉ & Chaoji Chen [2] ✉

In response to the looming concerns of plastic pollution, replacing plastic with paper is a very promising way, but its realization seems a long way off due to the poor water resistance and unsatisfied mechanical strength of cellulose fibril-based materials. Herein, we develop a versatile functionalizing material consisting of mainly biobased cyclic carbonate-bearing compounds and amine compound, which can enable the rapid transformation (within 2 min under microwave radiation) of the cellulose paper into plastic-like material (named paper plastic) having an unprecedently high tensile strength of ~126 MPa. Through a systematic experimental and theoretical study, the paper plastic's combination of excellent mechanical properties and water/solvent resistance is attributed to the easy formation of carbamate abundant non-isocyanate polyurethane cooperated with the intermolecular bond exchange mechanism between the dynamic carbamate moiety and hydroxyl of the cellulose. Also, benefiting from the high content (>80%) and natural advantages of biobased materials, the paper plastic shows significant thermal stability, processability, and biodegradability than most petrochemical-based plastics, promising the great potential of dynamic carbamate chemistry toward high-performing paper plastic composites.

Plastic products are becoming increasingly indispensable in people's daily lives with the development of modern society[1–3]. Meanwhile, the natural degradation of plastics takes hundreds or even thousands of years, posing grave environmental concerns[4–6]. Some panacea was sought to alleviate these complex environmental problems, such as using biobased raw materials or developing recyclable and degradable materials[7–10]. Paper composed of natural, renewable, and abundant cellulose has a long application history spanning millennia[11,12], so far, it still holds significant societal importance due to its merits of excellent lightweight nature, degradability, and recyclability[13,14]. In recent years, the implementation of the Plastic Ban has prompted consideration for using cellulose paper-based materials as substitutes for petrochemical-based plastics[15,16], which is, practically, challenged by the unsatisfactory mechanical strength and especially poor water resistance of cellulose paper[17–19]. The true realization of replacing plastic with cellulose paper urgently calls for approaches to simultaneously improve the mechanical properties and water resistance of cellulose paper, without sacrificing its degradability and recyclability[20].

Early efforts of compositing synthetic polymers with cellulose paper are considered a compromised choice due to the severe loss of degradability and poor recyclability, especially when thermoset polymers are used[21–23]. Alternatively, dynamic covalent polymers, with their topological networks capable of undergoing rapid rearrangement through the activation of dynamic covalent bonds[24–26], have attracted increasing attention due to their fascinating properties including recyclability, reprocessability, and self-healing ability, to serve as alternatives to traditional plastic materials[27–30]. On a deeper level, the worldwide goal of carbon neutrality[31], proposed in recent years, motivates the plastic research community to pay more attention to non-isocyanate polyurethane (NIPU) prepared by reacting carbonates with amino-containing compounds to form carbamate linkages. Worth noting is that NIPU can concurrently meet the property requirements and carbon neutrality goal when cyclic carbonates using $CO_2$ as a main starting chemical are involved[27,32–35]. Encouragingly, it was reported that the dynamic carbamate moiety would undergo bond cleavage and rebonding with hydroxy groups to form intermolecular cross-linkages[36]. Seychal et al.[37] used epoxy compounds or NIPU to composite with flax to obtain high-strength composites. In this work, it is confirmed that the strong hydrogen bond formed between NIPU and the fiber can promote the excellent longitudinally mechanical properties of the flax composite compared with the composite of epoxy resin and flax fiber, which is the result of the interaction between the carbamate bonds in NIPU and the hydroxyl group in the flax fiber. For cellulose paper that bears abundant hydroxy groups on the fibril surface, this dynamic carbamate chemistry is believed to provide vast unexplored opportunities for preparing advanced paper-plastic composites. This also provides a theoretical basis for us to exploit the NIPU modified cellulose paper.

Herein, we report a rationally designed ternary mixture composed of two biomass-derived cyclic carbonates and an amino-abundant polymer feedstock in certain ratios, capable of transforming cellulose papers into paper plastics possessing exceptional mechanical strength, water/solvent resistance, thermal stability, reprocessability, and degradability in minutes under microwave irradiation. Specifically, two carbonates, carbonated soybean oil (CSBO) and acrylpimaric acid cyclic carbonate (APAC), are prepared from $CO_2$ and natural product derivatives of epoxidized soybean oil (ESO) and acrylpimaric acid glycidyl ether (APADE), respectively. APAC with a rigid hydrogenated phenanthrene ring structure is introduced to further improve the mechanical performance. We experimentally and theoretically investigate the underlying strengthening mechanism of the obtained paper plastic and confirm that the formation of carbamate linkages and the dynamic bond cleavage–rebonding process between the carbamate moiety and hydroxy group of cellulose are primarily responsible for the high mechanical strength. Lastly, we demonstrate the generality and commercial viability of this transformative dynamic carbamate chemistry approach by continuously producing large-sized (up to 10 m in length) paper plastic films and ready-to-use straw, shopping bag, and drinking cup products from various (used) cellulose paper products.

## Results

### Fabrication and characterization of paper plastic

The fabrication strategy of paper plastic is schematically shown in Fig. 1a. The mixture solution consisting of CSBO, APAC, and PEI is coated onto the cellulose paper for sufficient impregnation and then subjected to microwave radiation-induced curing to obtain the paper plastic, in a continuous fashion. Chemically, both CSBO and APAC contain cyclic carbonate moieties, they can react with the amino groups of PEI to generate the non-isocyanate polyurethane (NIPU) with dynamic carbamate via a catalysis-free mechanism. The literature[36,38,39] confirms that the presence of hydroxyl groups is more conducive to the dynamic bond exchange reaction (transcarbamoylation) of

carbamate bonds (Supplementary Fig. 1a) in NIPU. Cellulose is known to be rich in hydroxyl functional groups. Therefore, the dynamic carbamate bond is capable of effectively undergoing bond exchange reactions with hydroxyl groups in cellulose paper (Supplementary Fig. 1b), making it an ideal choice for the preparation of paper plastics. Figure 1b presents the digital photographs of the cyclic carbonate solutions of CSBO and APAC, PEI solution, and their mixture solution. The synthetic routes of the cyclic carbonates are shown in Supplementary Figs. 2, 3. FTIR spectra (Supplementary Fig. 4) and NMR spectra (Supplementary Figs. 5, 6) confirm that the epoxides successfully react with $CO_2$ to obtain cyclic carbonate. With the ease of synthesis and incorporation of these cyclic carbonates into the paper substrates, the preparation of paper plastic could be easily scaled up (Fig. 1c). The corporation of NIPU and cellulose fibrils of paper, at both macro and micro levels, allows the as-prepared paper plastic to show a good combination of significantly improved mechanical properties, water resistance, thermal stability, degradability and recyclability, making it a sustainable alternative with broad applications over conventional plastics (Fig. 1d and Supplementary Fig. 7). In addition, the biobased carbon content of the paper plastic is estimated to be 82% according to the standard of ASTM D6866–22, demonstrating a close agreement with the theoretical value of 84.6% (Supplementary Note 2) for biobased carbon.

Figure 2a shows the design concept and possible reactions that the three initial reactants, CSBO, APAC, and PEI, would undergo during the preparation of paper plastics. A molar ratio of 1:1 between the cyclic carbonate groups (from CSBO and APAC) and the amino groups (from PEI) is employed to theoretically ensure the complete reaction of these functional groups to form carbamate dynamic bonds that are highly attractive for mechanical strength and bond exchangeability. To ascertain the proposed reaction route, a combined spectrum investigation was performed on the reactants, intermediates, and products. Comparing the FTIR spectrum of the obtained paper plastic with those of the reactants (Fig. 2b), it is seen that the absorption peak (1797 cm⁻¹) assigned to cyclic carbonate significantly weakens after the reaction, associated with the appearance of an absorption peak (1690 cm⁻¹) belonging to C=O of the carbamate bond. Meanwhile, the two absorption peaks in 3300–3500 cm⁻¹ belonging to –NH₂ of PEI disappeared. Collectively, the peak intensity evolution of these characteristic functional groups confirmed the synthesis route of NIPU. Moreover, original cellulose paper possesses a porous structure, which can facilitate the rapid penetration of the reactant mixture before curing. Consequently, the inter-fiber pores are completely and homogeneously filled during the curing process, as demonstrated by combined morphological and compositional analyses through the surface and cross-sectional SEM observation and the corresponding EDS elemental mapping images (N element exclusively attributable to NIPU) of the original cellulose paper and the as-prepared paper plastic (Fig. 2c and Supplementary Fig. 8). Subsequently, Raman and FTIR of cellulose and the cellulose plastic were performed to explore the carbamate bonding chemistry after the introduction of NIPU.

Comparing the Raman spectra of the cellulose paper[40] and the paper plastic, the absorption peak of C=O of carbamate is found at ~1600 cm⁻¹ with the incorporation of NIPU (Fig. 2d). The corresponding 2D Raman mapping (with an area of $200 \times 200\ \mu m^2$, red color represents a higher absorption intensity) further indicates the high abundance and uniform distribution of carbamate bonds across the paper plastic surface. In the FTIR spectrum of paper plastic (Fig. 2e), a typical carbonyl peak appears at 1740 cm⁻¹, accompanied by a –NH– bending vibration peak at 1550 cm⁻¹ corresponding to the free carbamate groups of the introduced polymer[41]. The formation of these groups occurs through the reaction between the amine groups and the cyclic carbonates. Furthermore, the presence of hydrogen bonds between carbamate groups or between carbamate groups and hydroxyl groups leads to the emergence of new absorption peaks in the range 1715 – 1695 cm⁻¹ for the paper plastic, compared with that of

**a**

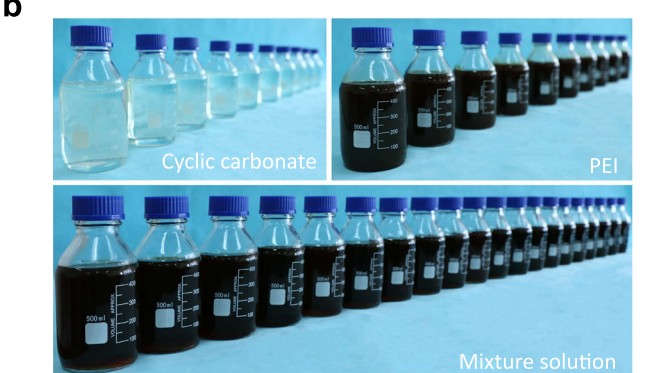

**b**

**c**

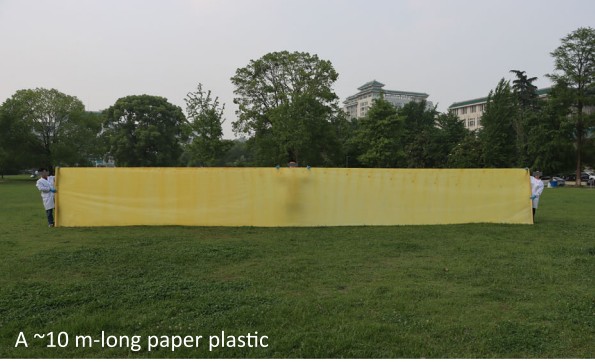

**d**

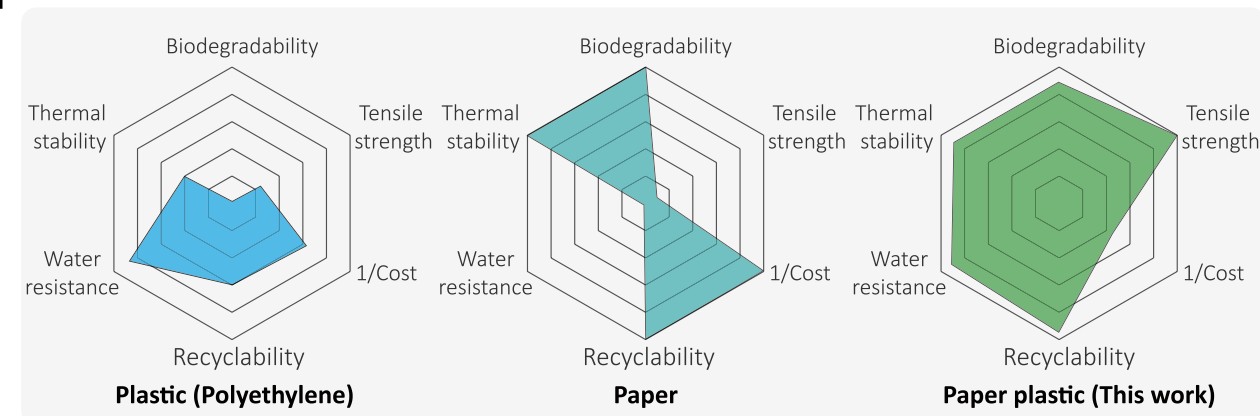

**Fig. 1 | Scalable, continuous, and rapid preparation of high-performance paper plastic based on a transformative microwave radiation-driven dynamic carbamate chemistry. a** Schematic demonstrating the preparation of the paper plastic. Firstly, the mixed solution of cyclic carbonate and PEI was compounded with cellulose paper, and then it was rapidly cured into paper plastic under microwave radiation. **b** Large-volume cyclic carbonates, PEI, and mixture solutions can be obtained. **c** A large-scale sheet of paper plastic. **d** Radar plots comparing the performance of paper plastic, cellulose paper, and polyethylene, in which the results are normalized by the maximum value of each characteristic. Source data are provided as a Source Data file.

the cellulose paper. The dynamic covalent bond can further undergo exchange reaction with −OH in cellulose paper (Fig. 2c), thus achieving efficient cross-linkage between NIPU and the cellulose substrate.

Additionally, we believe that during the microwaving-induced polymerization process, the carbamate moiety of NIPU would suffer from C−O bond cleavage and rearrangement reaction with the hydroxyl groups on the cellulose chain, based on the solid-state $^{13}$C NMR results (Supplementary Fig. 10): in addition to the new peak at 153.7 ppm attribute to the carbamate group[42], the peak shift of C1 (107.3 ppm) and C4 (90.8 ppm) of cellulose[43] to the high field and a significant shoulder peak of C6 (65.4 ppm) were observed, which is attributed to the cross-linking between the C6−OH and NIPU. Consistently, theoretical calculation reveals that the formation of

carbamate connection between NIPU and cellulose is thermodynamically favorable with a high binding energy (Supplementary Table 2).

## Mechanistic insights into the strengthening mechanism of paper plastic

The mechanical properties of common cellulose paper are not competitive enough compared to plastic materials. Moreover, cellulose paper is more prone to breakage upon moisture exposure or being wetted due to the sharp decrease of contact areas between cellulose fibers and hence the number of hydrogen bonds in these areas. As shown in the tensile strain−stress curve, the cellulose paper used in this work shows a fracture strength of 13 MPa, with the incorporation of NIPU, a drastic

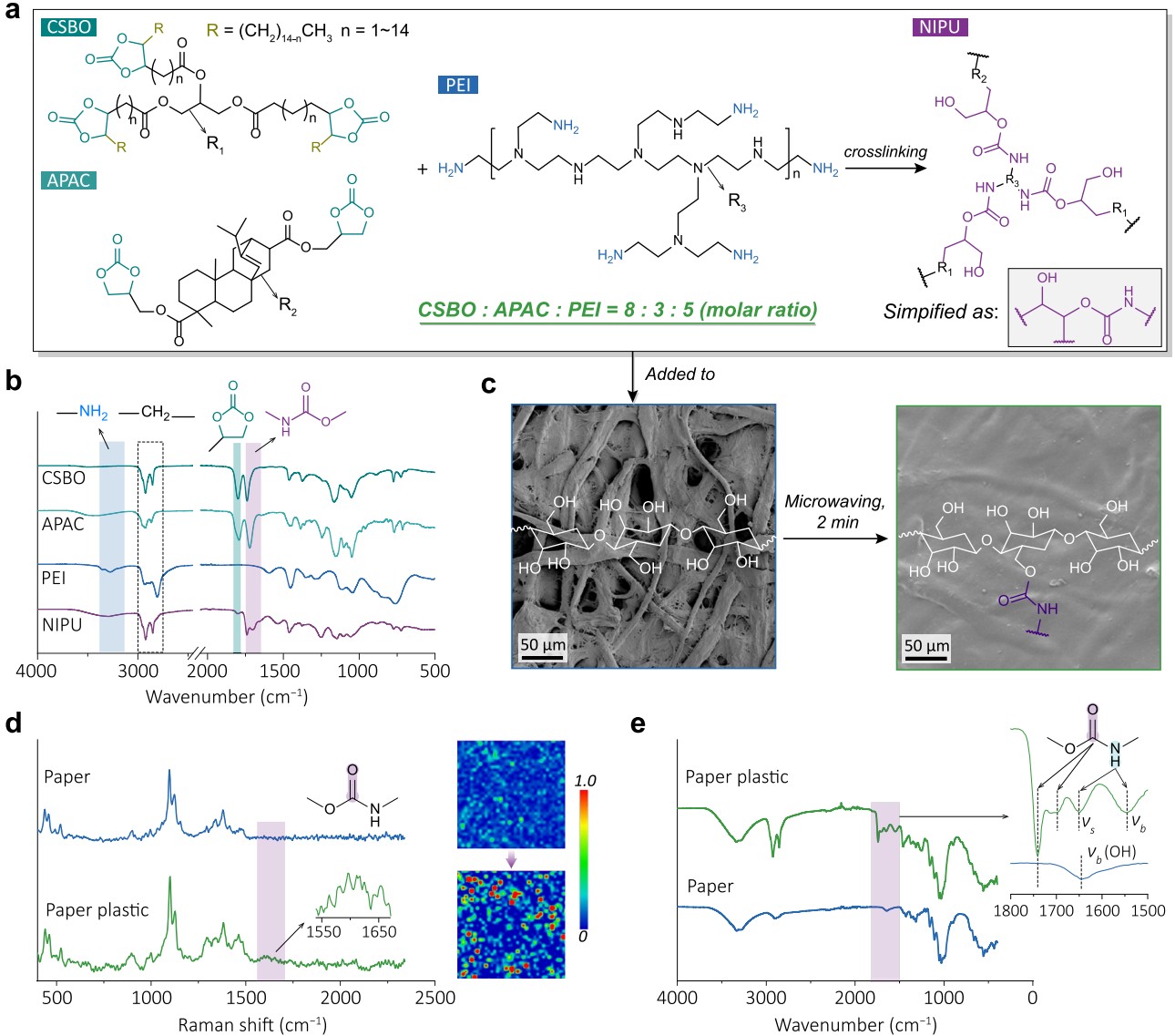

**Fig. 2 | Morphological and structural features of paper plastic. a** Schematic of polymer (NIPU) preparation. **b** FTIR spectra of CSBO, APAC, PEI, and NIPU. **c** SEM images of cellulose paper and paper plastic. Scale bar, 50 μm. **d** One-dimensional Raman spectra and two-dimensional (2D) Raman spectra constructed from the intensity of C=O. **e** FTIR spectra of paper and paper plastic. Source data are provided as a Source Data file.

improvement, by a factor of ~10, to 126 MPa was observed for the paper plastic (Fig. 3a). Concurrently, 2.5 and 24 times increase of the Young's modulus and toughness compared to the cellulose paper were also achieved (Fig. 3b). In a more visual demonstration of such exceptional mechanical properties of the as-prepared paper plastic, a paper plastic sample measuring 100 mm (width) × 0.36 mm (thickness) was shown to support an adult weighing 65 kg (Supplementary Fig. 13). The combination of these mechanical properties makes the paper plastic a promising alternative to numerous natural and synthetic materials (Fig. 3c). Considering the low tensile strength of only ~3.2 MPa for NIPU, it is reasoned that the reformation of hydrogen bonds[37] and carbamate linkages between cellulose and NIPU is responsible for the significant interfacial reinforcement and therefore the mechanical properties.

To understand the microstructural evolution and performance improvement in paper plastic, we constructed multiscale simulations and analyses for the paper plastic based on experimental evidence (see Supplementary Information for details). The simulated strain–stress curves exhibit good agreement with the experimental measurements. The simulated stresses corresponding to the experimental failure strains for cellulose paper and paper plastics are 12 and 156 MPa,

respectively, which align with the experimental values (13 and 126 MPa, respectively). To demonstrate the decisive role of carbamate linkage between cellulose and NIPU in boosting the mechanical performance, an elastic silicone rubber (SR) that shows very similar mechanical behavior (rubber-like) and tensile strength (~3 MPa for NIPU vs. 1.38 MPa for SR) with NIPU, and meanwhile is unlikely to form covalent cross-linking, was synthesized to serve as a control in this study. As the morphological, compositional, and mechanical analyses reveal, although the cellulose paper can also be homogeneously and fully filled the SR, a tensile strength of only 21 MPa was measured (Supplementary Figs. 15, 16). This indirectly but strongly suggests that chemical cross-linking between cellulose and NIPU did occur within the paper plastic. Furthermore, the microstructural evolutions of microfibrils in cellulose paper and paper plastic under uniaxial tension were analyzed. Upon a certain stretching, the stress levels of paper plastic are significantly higher than cellulose paper (Fig. 3d, e and Supplementary Figs. 17–19), which is attributed to that the carbamate linkage improves stress transfer efficiency. Our simulations show that the existence of a carbamate linkage can significantly impede the slippage of microfibrils, thus enabling exceptional mechanical properties of the

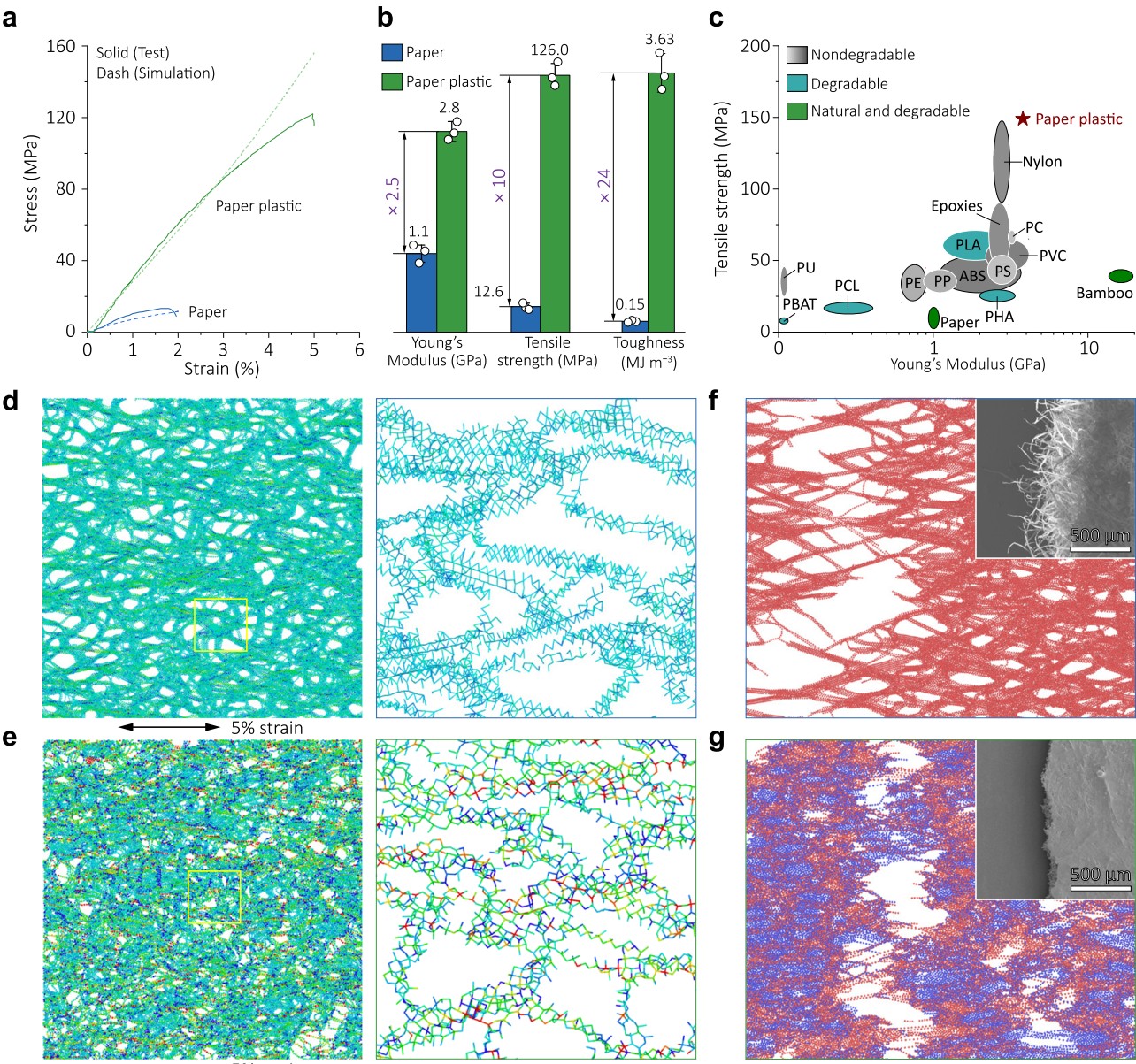

**Fig. 3 | Enhanced mechanical properties and mechanistic insights into the strengthening mechanism of paper plastic. a** Stress–strain curves of the cellulose paper and paper plastic. **b** Comparison of Young's modulus, tensile strength, and toughness of paper (Young's modulus: 1 GPa, strength: 13 MPa, toughness: 0.15 MJ/m³) and paper plastic (Young's modulus: 3 GPa strength: 126 MPa, toughness: 3.63 MJ/m³). Values in b represent their means ± SDs from $n$ (number of the repeated experiment) = 3 independent samples. **c** Ashby diagram of tensile strength vs. Young's modulus for paper plastic compared with typical polymers. **d** Color-coded two-dimensional maps of stresses carried by cellulose paper at a strain of 5%. **e** Color-coded two-dimensional maps of stresses carried by paper plastic at a strain of 5%. **f** SEM image and simulated fracture surface features of the paper. Scale bar, 500 μm. **g** SEM image and simulated fracture surface features of paper plastic. Scale bar, 500 μm. Source data are provided as a Source Data file.

paper plastic. These simulations are also supported by the experimental observations, for example, both SEM images and simulations show that a flat fracture surface is retained for the paper plastic (Fig. 3f, g and Supplementary Fig. 20), while the cellulose paper shows a messy fracture surface due to the slippage of cellulose microfibrils.

Considering the feasibility and high efficiency of this dynamic carbamate chemistry approach, it is theoretically applied to various types of daily-used paper products with cellulose materials as the primary component. Using the same process parameters, paper plastics based on a wide spectrum of cellulose paper products including hardwood pulp paper (HPP), softwood pulp paper (SPP), filter paper (FP), kraft paper (KP), printer paper (PP), tissue paper (TP), and newspaper (NP) were prepared with their digital photographs shown in

Supplementary Fig. 21. The tensile stress–strain results reveal that after compounding with NIPU the mechanical properties of all cellulose papers can be significantly improved by almost one order of magnitude (Supplementary Fig. 22). Taking HPP, SPP, and SPP as examples, their tensile strengths are determined to be 72.95, 94.26, and 103.16 MPa, respectively, to list a few. These results substantiate the applicability and generality of the preparation method for high-performance paper plastics.

**Other required properties for plastic substitution applications**
Conventional plastic materials are usually hard to operate under extreme temperature environments due to their insufficient thermal dimensional stability[4,44]. The limitation is well addressed for our paper

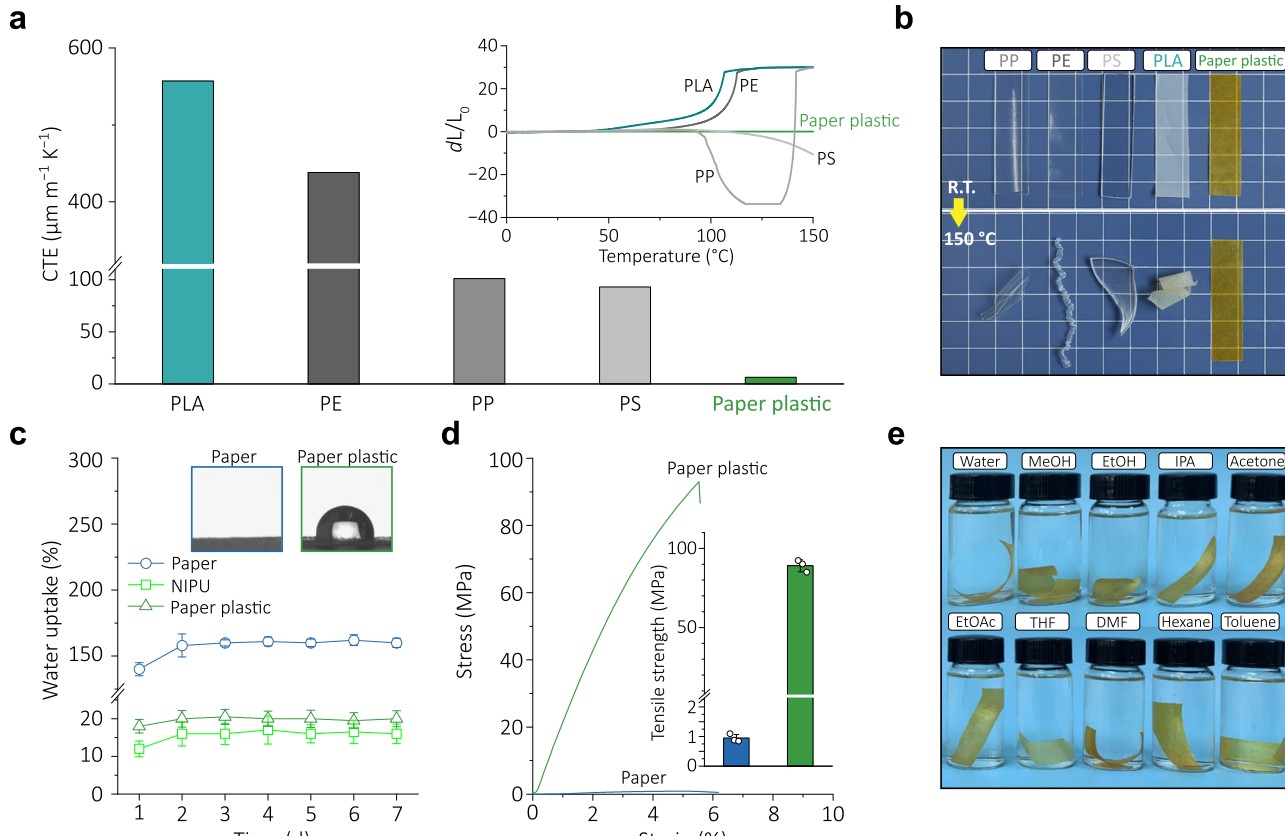

**Fig. 4 | Thermal, water, and solvent stability of paper plastic. a** Coefficient of thermal expansion (CTE) and thermal expansion of polypropylene (PP), polyethylene (PE), polystyrene (PS), poly(lactic acid) (PLA), and paper plastic. **b** Thermal stability experiment. Comparison of the paper plastic with widely used petroleum-based plastics at 27 °C (RT) and 150 °C. Compared to 27 °C, petroleum-based plastics have already fully softened and deformed at 150 °C, while paper plastic still shows no visible change. **c** Water uptake of cellulose paper, NIPU, and paper plastic. The error bars represent the standard errors from three samples for each material. The insets are photographs for water contact angle measurement for the cellulose paper and paper plastic samples. **d** Tensile strain–stress curves of cellulose paper and paper plastic after soaking in water for 7 d. The inset shows the measured tensile strength with error bars representing the ±SDs (number of the repeated experiment = 3). **e** Stability of paper plastic in organic solvents after 7 d. Source data are provided as a Source Data file.

plastics, mainly benefiting from the excellent thermal (dimensional) stability of the cellulose paper, thereby making them more competitive in plastic substitution. Quantitatively, according to the coefficient of thermal expansion (CTE) test results (Fig. 4a), the paper plastic exhibits a rather low CTE value of 3.54 μm/(m·°C) (determined in the temperature range 0–100 °C), significantly lower than widely used commercial plastics (PP, PE, PS, and PLA). More intuitively, commercial plastics exhibit softening and shrinkage when placed on a hot plate at 150 °C for 20 min, whereas the paper plastics have no visible changes (Fig. 4b). For better comparison with commercial plastics, we have also supplemented the thermal expansion test results of epoxy and polyurethane (PU) (Supplementary Fig. 23). These results show that paper plastics can exhibit thermal stability similar to thermosetting epoxy resins, and further prove that paper plastics have well thermal stability. In addition, TGA testing showed that the initial weight loss temperature ($T_{5\%}$) of paper plastic exceeded 250 °C (Supplementary Fig. 24), indicating the excellent thermal stability of the paper plastic. The TG–FTIR test also confirmed the significantly suppressed thermal decomposition process of the paper plastic, as shown in Supplementary Fig. 25.

The hydrophilicity of cellulose often leads to inadequate water resistance for paper packaging materials[45,46], necessitating the urgent need for addressing this issue and enhancing their water resistance. By incorporating NIPU with hydrophobic aliphatic chains to encapsulate the hydrophilic cellulose paper fibers, the paper plastics exhibit satisfactory waterproof ability. The NIPU-coated paper exhibits a water

contact angle of over 100° (Fig. 4c), demonstrating significant hydrophobicity compared to unmodified paper (almost no contact angle). Taking into account the influence of hydroxyl groups in the paper plastic, we supplemented the evolution of the contact angle over time. As shown in Supplementary Fig. 26b, the contact angle of the paper plastic surface hardly changes as the time extends from 0 to 120 min, and remains at about 100°, which indicates that the stable hydrophobicity and suggests that the crosslinked NIPU network effectively restricts water penetration. Moreover, cellulose paper and paper plastics are immersed in water for 7 days, and their water absorption is recorded. Figure 4c shows that cellulose paper exhibits 160% water absorption, while paper plastic absorbs only 20%. The wet tensile strength of the paper plastic can be maintained at ~89 MPa (about 72% of the dry paper plastic) after soaking for one week (Fig. 4d). These results indicated that the incorporation of NIPU into the cellulose paper network effectively enhances the mechanical properties of paper plastic in wet environments. Importantly, this hydrophobicity is achieved while maintaining high biobased content (82%) and full recyclability, which represents a key advance over conventional petroleum-based plastic coatings. Furthermore, Paper plastics were immersed in various organic solvents (MeOH, EtOH, IPA, acetone, EtOAc, THF, DMF, hexane, and toluene) to study their solvent resistance (Fig. 4e). After 7-day immersion, no noticeable color and size change for the paper plastic was observed (Supplementary Fig. 26c), showing excellent solvent resistance across a wide variety of common solvents.

## From raw materials to products

Because the dynamic carbamate linkage can break and reform with moderate energy input, paper plastic components can readily be bonded together to prepare various products. In a simple attempt, single-layer paper plastics were effectively combined through hot pressing to obtain high-strength boards (Fig. 5a). SEM image analysis (Supplementary Fig. 27a) reveals a closely stacked cross-section architecture of the multi-layer paper plastic after hot pressing. Supplementary Fig. 28a demonstrates the load-bearing capacity of the paper plastic board. Figure 5b shows that the edges of the paper plastics can be hot pressed to realize heat-sealing through the dynamic bond exchange reaction for the demonstration of daily packing bags (see Supplementary Fig. 29 for more details). To examine the adhesion limit, two rectangular paper plastics were subjected to a tensile test on a universal testing machine after being welded by hot pressing to verify their heat-sealing ability. It is evident that the paper plastic fractures during the test, while the welded position remains intact (Supplementary Fig. 30), which demonstrates its excellent heat-sealing ability for adhesive-free packaging products. Given the excellent water resistance of the paper plastic, cups and straws were produced and tested, with commercial products as the control group, unexpectedly, only the paper plastic group exhibits persistent wet strength, allowing continuous use in various aqueous environments for over 6 h (Supplementary Fig. 31).

Compression and Charpy pendulum tests were performed[47] to demonstrate the mechanical properties of the paper plastic board. The compressive strength of a paper plastic board is 96.5 MPa (Fig. 5c), while its impact toughness measures 170.6 kJ/m$^2$ (Fig. 5d), surpassing that of a considerable number of commercial plastics. These results suggest that the paper plastic possesses good thermal lamina ability due to the presence of dynamic covalent bonds, thereby enabling its application as building blocks of composite board. Additionally, the paper plastics can be easily imbued with oily dyes (Supplementary Fig. 32), thereby augmenting the chromatic diversity. The gas barrier is also an important property of packaging materials[48]. Figure 5e and Supplementary Fig. 33a demonstrate significantly lower oxygen transmission rate (OTR, 72 cm$^3$/(m$^2$·24 h·0.1 MPa)) and nitrogen transmission rate (NTR, 25 cm$^3$/(m$^2$·24 h·0.1 MPa)) for paper plastic compared to cellulose paper alone. The reason is that NIPU can penetrate the original pores between the paper fibers, blocking the passage of gas in the cellulose paper. Additionally, the formation of a NIPU film on the surface of hydrophilic paper fiber further hinders gas diffusion. Moreover, the paper plastic shows desirable gas barrier (indicated by oxygen transmission rate) comparable and modest moisture barrier (indicated by water vapor transmission rate) abilities compared to commercial plastics[49], further validating its application potential as packaging materials (Supplementary Fig. 33b).

## Biocompatibility analysis of paper plastic

To further broaden the application properties of paper plastics, biocompatibility experiments were carried out, including in vitro cytotoxicity test, skin irritation test, and skin sensitization test. After cultured with paper plastic extract and L-929 cells for 24 h, the cell morphology was observed, and the cell viability ratio was quantified by the MTT method. As shown in Fig. 5f and Supplementary Fig. 34, the cell viability ratios of 25, 50, 75, and 100% sample extracts were all higher than 80%, while maintaining predominantly intact cellular morphology. These results confirm that the paper plastic has less cytotoxicity to L-929 cells and meets the acceptance standard according to the national standard ISO10993−5: 2009. In addition, skin irritation and sensitization tests are also carried out as the materials will direct contact with human skin. Following the ISO10993−10: 2010 standard, paper plastic was extracted with 0.9% NaCl injection (polar extract) and sesame oil (non-polar extract). Then, gauzes

moistened with the extracts were applied to the back of the New Zealand rabbit for a skin irritation experiment, which were removed after 4 h. The animals were observed at 24, 48, and 72 h after the gauze was removed. Figure 5g illustrates no presence of erythema or edema on the skin surface. Meanwhile, based on the classification system of skin reactions (Supplementary Tables 3, 4), all experimental rabbits exhibited a primary irritation index of 0 when exposed to paper plastic (Supplementary Table 5), indicating its non-inductive nature towards skin irritation. The skin sensitization test was conducted using the guinea pig maximization test (ISO10993−10: 2010 standard, Fig. 5h). Firstly, the extract was injected into the medial scapula of guinea pigs for intradermal induction, and then the gauzes wetted by the extract was applied to the injection site of the animal for 48 h to experience topical induction after 7 days. After 15 days, the gauzes wetted by the extract were applied to the left and right abdomen of guinea pigs for 24 h to challenge. The appearance of the injured part of the animal was observed 24 and 48 h after the removal of the dressing, revealing no signs of erythema and edema (Fig. 5h). Moreover, according to the Magnusson and Kligman scale provided in Supplementary Table 6, paper plastic exhibited a sensitization grade score of 0, indicating a sensitization positive rate of paper plastic at 0%. Concurrently, positive control and negative control experiments were conducted on experimental animals to confirm the effectiveness of paper plastic skin irritation and sensitization tests.

## Environmental impacts of paper plastic

Cellulose paper is a highly biodegradable material. Therefore, it is desirable to retain its inherent ability to degrade easily even after modification. To this end, the cellulose paper, paper plastic, NIPU, and commercial plastics (polyethylene (PE)) were buried in a natural environment (Nanjing, China) for performing biodegradability tests (Fig. 6a and Supplementary Fig. 35). The NIPU was introduced into the cellulose paper is derived from biobased raw materials (APAC and CSBO) and biodegradable PEI[50], enabling the paper plastic to undergo decomposition by microorganisms and participate in the cycle of the ecological environment after being buried in the soil. Similar to the rapid degradation of cellulose paper observed in soil, the paper plastic exhibited initial signs of degradation after being buried in the soil for 1 month, as evidenced by the presence of holes on its surface, which could be attributed to microbial activity and enzymatic action in the soil[23,51]. Subsequently, an increase in microbial content on the material surface facilitated the hydrolysis of hydrogen bonds and chemical bonds[27] in the paper plastic, which promoted the complete degradation of the paper plastic within 180 days. Meanwhile, noticeable degradation of NIPU was observed at 210 days. In contrast, the commercial PE buried in the soil exhibits no changes and may require several decades to centuries for degradation. Paper plastics can maintain stability under normal working conditions but undergo degradation when buried in soil, highlighting their exceptional biodegradability. In addition, data on the amount of $CO_2$ produced by microorganisms during the biodegradability process (Supplementary Fig. 36) and the possibility of microplastics being produced (Supplementary Fig. 37) after the degradation of paper plastics are measured. The results confirm that the paper plastics are biodegraded during the degradation process, and there is no obvious production of microplastics. This characteristic renders it a promising alternative to conventional plastic packaging materials.

Due to the dynamic carbamate bond constructed in paper plastics, they can also be recycled in addition to experience biodegradation, which challenges the problem of traditional plastics that are difficult to degrade and effectively alleviates the environmental problems. The carbamate groups containing β−OH formed after introducing NIPU into the cellulose paper can undergo a bond exchange reaction with the hydroxyl groups. This enables the rearrangement of the network structure and facilitates the reprocessing of paper

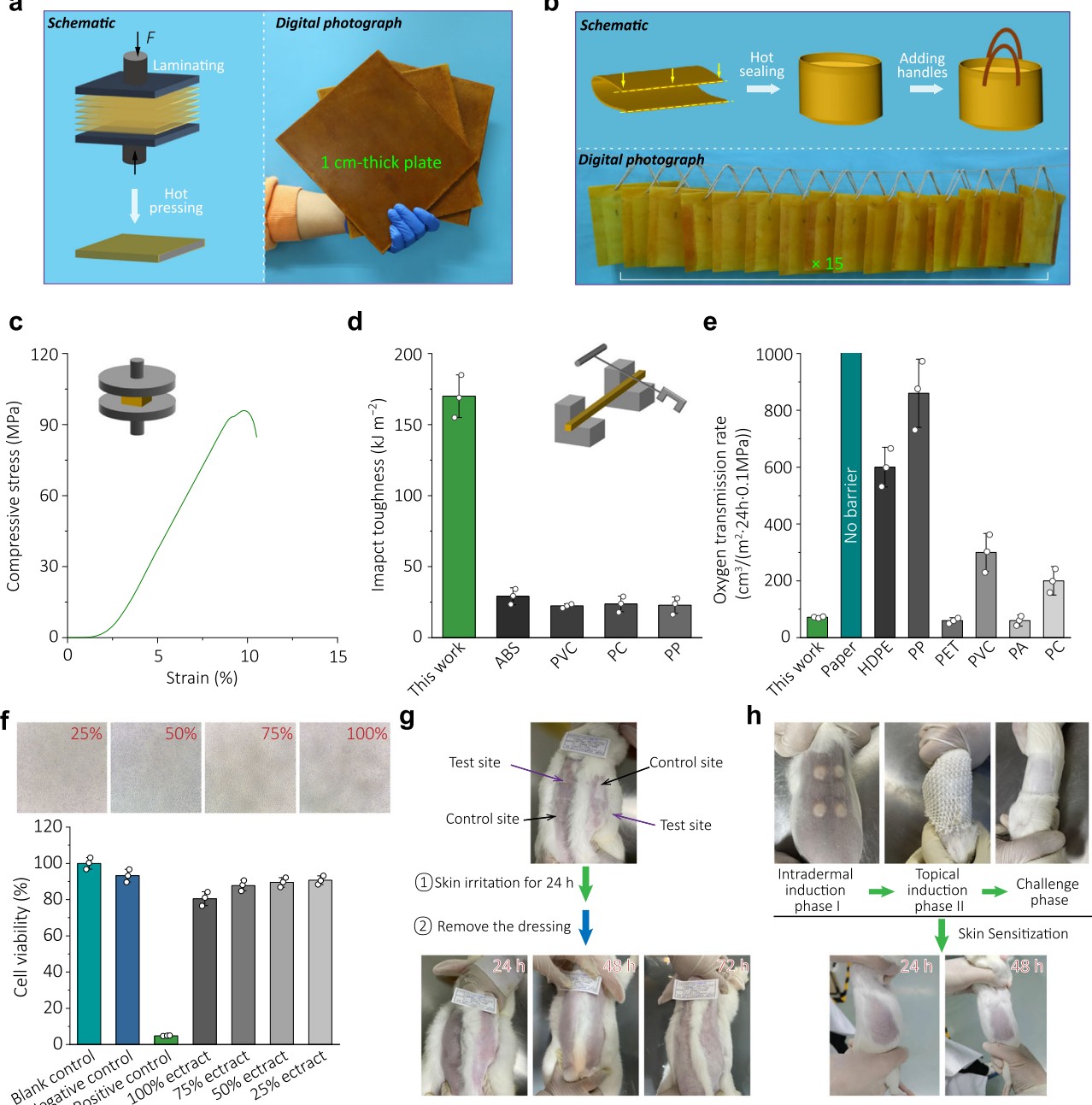

**Fig. 5 | Production of various paper plastic products with excellent comprehensive performances. a** Schematic illustration showing the preparation process of thick paperboard from multiple thin paper plastic sheets by hot pressing and the photographs of multiple paper plastic boards. **b** Schematic illustration of the plastic bag preparation process using paper plastic. A large number of bags are made of paper plastic. **c** Compression stress–strain curve of paperboard. **d** Comparison of Charpy impact toughness of paper plastic with other widely used polymer-based materials. **e** The oxygen transmission rate of different plastics and paper plastic at 23 °C/90% RH. HDPE high density polyethylene, PP polypropylene, PET polyethylene terephthalate, PVC polyvinyl chloride, PA polyamide, PC polycarbonate, ABS acrylonitrile butadiene styrene. **f** In vitro cytotoxicity test of paper plastic, including cell morphology and cell viability ratios of 25, 50, 75, and 100% sample extracts. Error bars in **d**, **e**, **f** indicate ± SDs from three measurements. **g** Skin irritation test of paper plastic. **h** Skin sensitization test of paper plastic. Source data are provided as a Source Data file.

plastics. As shown in Fig. 6b, the fragmented paper plastic can become a complete material after hot pressing. However, due to the presence of non-malleable cellulose paper components comprising 50% of the recycled content, discernible surface textures are observed on the reprocessed paper plastic. Nevertheless, as evidenced by the SEM image of the cross-section, the re-cross-linking process has successfully taken place. Moreover, Fig. 6c shows the chemical recyclability of paper plastics. The carbamate bonds in the paper plastics readily undergo alkaline hydrolysis when exposed to boiling NaOH solution,

resulting in the depolymerization of NIPU into smaller molecules and separation from the cellulose paper substrate. The cellulose paper fibers can be easily collected for recycling through filtration and subsequently reconstituted into a complete piece of paper after the papermaking process. This straightforward approach demonstrates a simple yet effective method for achieving the recycling of paper plastics. Moreover, the recovery experiments of the pure polymer in the paper plastic (NIPU) were provided to demonstrate dynamic carbamate chemistry more clearly (Supplementary Fig. 38). The re-curing

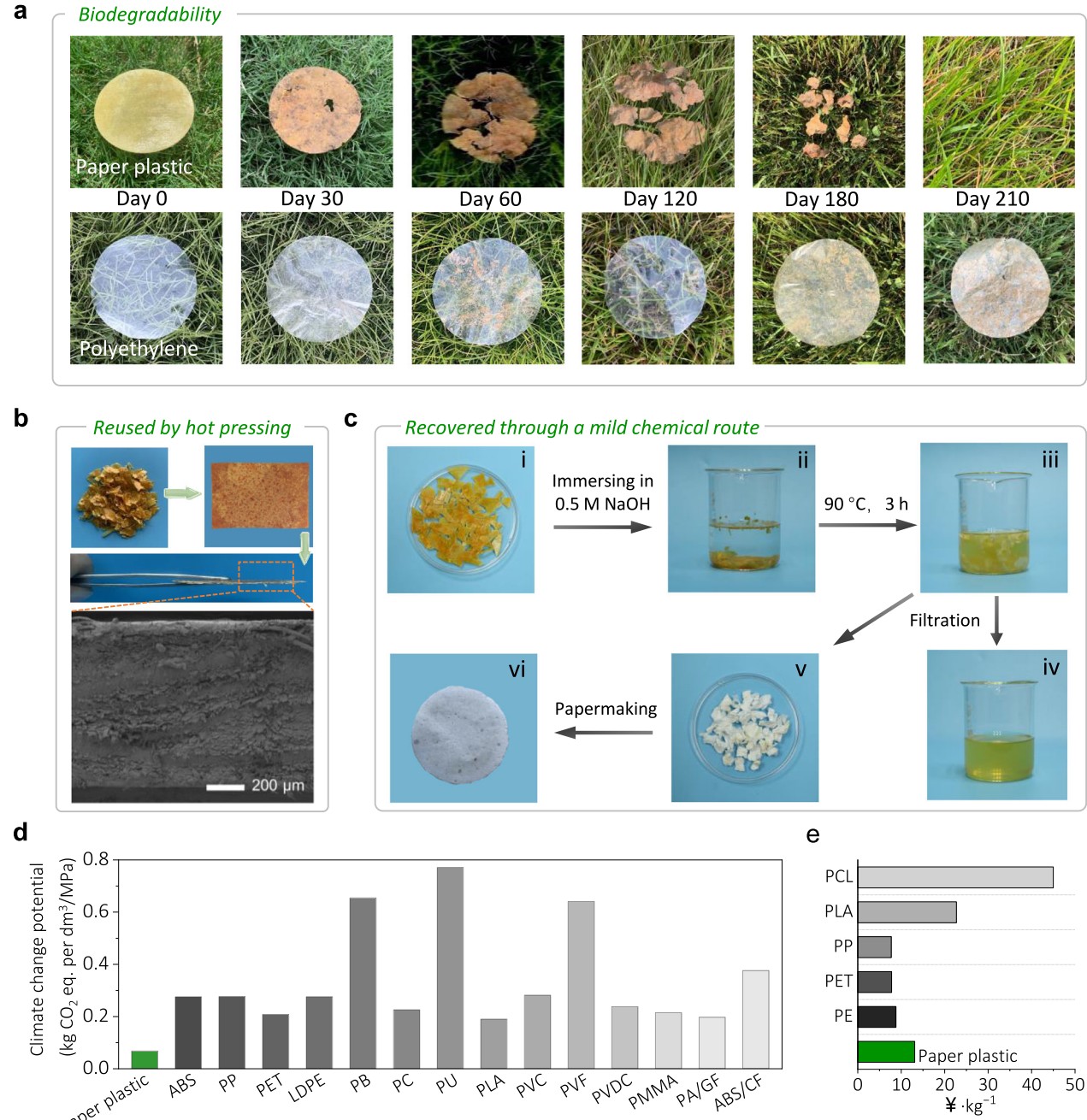

**Fig. 6 | Environmental impacts of paper plastic. a** Degradation process of paper plastic, and polyethylene (PE) buried in soil (Nanjing). **b** Physical recyclability of paper plastics. Shredded paper plastic recovered to intact paper plastic after hot pressing and cross-section SEM image of the reprocessed paper plastic. Scale bar, 200 μm. **c** Chemical recyclability of paper plastics. Photos show the degradation of paper plastic in 0.5 M NaOH solution and the regeneration of cellulose paper from recyclables. **d** Cradle-to-grave climate change potential (normalized to tensile strength) of the paper plastic compared to 14 potentially competitive benchmark plastic films and composite materials using life cycle assessment (LCA). **e** Comparison of the economic cost of paper plastic to commercial plastics (PE, PET, PP, PLA, and PCL) via TEA. PP polypropylene, LDPE low density polyethylene, PB polybutylene, PU polyurethane, PLA polylactic acid, PVF poly(vinyl formal), PVDC polyvinylidene chloride, PMMA poly(methyl methacrylate), PA/GF polyamide/ fiberglass, ABS/CF acrylonitrile butadiene styrene/carbon fiber, PCL polycaprolactone, PE polyethylene. Source data are provided as a Source Data file.

NIPUs obtained by physical hot pressing and chemical degradation confirm the dynamic properties of the network structure.

Life cycle assessment (LCA) was applied to determine the cradle-to-grave environmental impacts of paper plastic, i.e., from resource extraction to end-of-life (EoL). The climate change potential is used to understand the environmental sustainability of the material because it quantifies the greenhouse gas emissions driving climate change, a major global challenge nowadays. Besides, it also provides a common reference for comparing with other products or processes. The climate change potential (normalized to tensile strength as done by ref. 6) in Fig. 6d shows a significantly lower value for paper plastic over 14 potentially competitive benchmark plastic films and composite materials. Such a notably low carbon footprint originates from the manufacturing process simplicity, significant biogenic carbon content, and high strength of the paper plastic. As shown in Supplementary Fig. 39, the manufacturing of paper plastic (cradle-to-gate) presents a carbon footprint of 6.81 kg $CO_2$ equiv kg$^{-1}$, which is just under average for conventional plastics and composites. As a matter of fact, the

obtained footprint value is just slightly above the cradle-to-gate carbon footprint of 2.6 and 2.9 kg $CO_2$ equiv $kg^{-1}$ of HDPE and LDPE[52], or the -4.75 kg $CO_2$ equiv $kg^{-1}$ reported by ref. 53 for non-isocyanate poly-thiourethane production (no biogenic credits), a sustainable alternative to conventional plastics. However, it should be noted that the obtained footprint represents a conservative value, as the PEI modeled has a carbon footprint of 21.74 (lower values are more likely for industrially processed PEI). Furthermore, the biodegradability and large biogenic content of the paper plastic are translated into reduced $CO_2$ emission at the EoL. The incineration of conventional plastics and composite materials is subjected to additional burdens because the additional energy input for burning plastics, as well as for the $CO_2$ emissions originating from the fossil carbon that remains in the plastics (this adds approximately 6 kg $CO_2$ equiv $kg^{-1}$). On the contrary, materials from biobased feedstocks can be composted and the $CO_2$ formed during aerobic composting can be accounted as burden free because it will be included in the short-term carbon cycle again[54]. The footprint per kilogram of final material is therefore lower, which together with its good mechanical properties makes paper plastic a low carbon material when the tensile strength is taken as a functional unit (as shown in Fig. 6d). As shown in Supplementary Fig. 40, the environmental impact of paper plastic is primarily attributed to PEI, which contributes 11.5 to 54.6% of the impact, ranking first in 13 out of the 16 environmental impact categories. Cellulose paper, soybean oil and acrylpimaric acid are the other major contributors, especially in the categories of land use, marine eutrophication and non-carcinogenic human toxicity, respectively. Unlike other studies, the energy consumption in paper plastic production is very low, with a maximum contribution of 0.1%.

As a green and environmentally friendly material, the techno-economic analysis (TEA) of preparing paper plastics is also crucial. Therefore, we calculate the cost of paper plastics to assess their technical economics (Fig. 6e). Although the production cost of paper plastics (13.1 ¥/kg) is higher than that of some petrochemical-based plastics, such as PE (8.8 ¥/kg), PET (7.8 ¥/kg) and PP (7.7 ¥/kg), paper plastics have better environmental friendliness. Meanwhile, the cost of paper plastics is better than that of degradable plastics PLA (22.7 ¥/kg) and PCL (45 ¥/kg). These results indicate that the preparation strategy of paper plastics is environmentally friendly and economically feasible, which improves the development of the industry by replacing plastics with paper, and also contributes to emission reduction and economic impact.

## Discussion

In general, we employed a streamlined and efficient assembly line process to fabricate an innovative paper plastic possessing exceptional strength, water resistance, thermal stability, biodegradability, and recyclability. The process entails blending cyclic carbonates with amines and compounding the mixture with cellulose paper, followed by microwave radiation-driven curing to produce paper plastics. The carbamate bond formed by the reaction between cyclic carbonate and amine was exchanged with the hydroxyl group in the cellulose paper to achieve stable chemical cross-linking. This method effectively improves the mechanical strength of the cellulose paper, demonstrating that the tensile strength of the paper plastic is greater than 120 MPa, which is nearly tenfold higher than that of the cellulose paper. The incorporation of the hydrophobic segments enhances the water resistance, as evidenced by a tensile strength exceeding 80 MPa even after immersion in water for one week. Paper plastics exhibit stable dimensions and thermal resistance over a wide temperature range due to the exceptional thermal stability of cellulose. Moreover, benefiting from environmentally friendly raw materials, paper plastics still display biodegradability comparable to that of cellulose paper. The incorporation of dynamic carbamate bonds endows the paper plastic with excellent chemical degradation and physical reprocessing ability. This

process enables the rapid and continuous production of paper plastics on a large scale, and their sustainability and exceptional performance render them potential substitutes for conventional plastics.

## Methods

### Materials and chemicals

Spruce pulp (softwood pulp) and eucalyptus pulp (hardwood pulp) were purchased from Qingdao Jiunuo International Trading Co., Ltd. Softwood pulp papers (SPP) and hardwood pulp papers (HPP) were prepared by laboratory papermaking equipment. The mixed pulp paper (MPP) was prepared by papermaking equipment after mixing 70% softwood pulp and 30% hardwood pulp. The filter paper (medium-rate qualitative, FP) was purchased from Hangzhou Wohua Filter Paper Co., Ltd. The newspaper (NP) is Jinling Evening News. Kraft paper (KP) and printing paper were purchased from Deli Group Co., Ltd. Acryl-pimaric acid (APA, 95%) was provided by Wuzhou Sun Shine Forestry & Chemicals Co., Ltd. Epichlorohydrin (99%), Epoxidized soybean oil (ESO, 98%, epoxy value ≥6), L-ascorbic acid (99%), tetra-butylammonium iodide (TBAI, 99%), and polyethyleneimine (PEI, ≥99%, $M_w$ = 1800) were obtained from Macklin Inc. Methanol (MeOH, 99%), ethanol (EtOH, 99%), isopropanol (IPA, 99%), acetone (99%), ethyl acetate (EtOAc, 99%), tetrahydrofuran (THF, 99%), N, N-dime-thylformamide (DMF, 99%), hexane (99%), sodium hydroxide (NaOH, ≥97%) and toluene were provided by Nanjing chemical reagent Co., Ltd. The water used in the experiment is deionized water. All the materials and chemical reagents were used without further purification.

### Fabrication of cyclic carbonate

CSBO was obtained by adding 100 g ESO, 1.66 g TBAI, and 0.79 g L-ascorbic acid into the pressure reactor, then filling $CO_2$ to a pressure of the reactor up to 0.3 MPa and reacting at 80 °C for 24 h. The preparation of APAC first needs to obtain acrylpimaric acid glycidyl ether (APADE). About 100 g APA and 21.3 g NaOH were dissolved in 494 g epichlorohydrin and reacted at 117 °C for 4 h, then the excess epi-chlorohydrin was recovered by rotary distillation to obtain acrylpi-maric acid diglycidyl ether. Furthermore, 100 g of acrylpimaric acid diglycidyl ether, 1.52 g of TBAI, and 0.72 g of L-ascorbic acid were added to the reactor to react at 80 °C for 24 h with a $CO_2$ pressure of 0.3 MPa for obtaining the acrylpimaric acid cyclic carbonate (APAC). Supplementary Figs. 1, 2 exhibit the preparation process of CSBO and APAC. The cyclic carbonate compounds obtained in the high-pressure reactor are directly collected and applied to the subsequent paper plastic preparation process, except for the residue in the reactor chamber, the rest are all products.

### Fabrication of paper plastic

About 50 g of APAC, 310 g of CSBO, and 140 g of PEI were first stirred at room temperature for 30 min to obtain a homogeneous mixture. During the mixing of the raw cyclic carbonate compounds with the amine compounds, the temperature of the mixing system is almost stable around room temperature (about 25 °C) within the mixing process. After about 30 min of mixing, the viscosity of the mixed system increases to 4000 mPa·s, at which point the mixture is applied to the surface of the cellulose paper (500 g, 60 g/m²) with a roller brush and cured through microwave radiation (6 kW, Nanjing Hurui Micro-wave Technology Development Co., Ltd) for 2 min to prepare paper plastic. Paper plastics were obtained directly using unpurified cyclic carbonate. Meanwhile, the catalytic system would decompose to produce tributylamine, which might catalyze the ring opening of the cyclic carbonates and the formation of carbamate bonds[55]. Paper plastic board and paper bag products are prepared by stacking paper plastics on top of each other and then pressing them at 120 °C and 5 MPa for 1 h.

## Characterization

Fourier transform infrared (FTIR) spectra of different samples were performed with a Nicolet iS50 FTIR spectrometer (Thermo Fisher). Solid-state $^{13}C$ NMR measurements were conducted on an NMR spectrometer (Bruker Advance NEO, 500 MHz)[1].H NMR spectra were recorded on a Bruker Avance III 400 MHz, respectively. X-ray diffraction (XRD) patterns of the samples were collected with a Siemens D5000 X-ray diffractometer (Siemens, Germany) at a scanning speed of 10°/min. Scanning electron microscopy (SEM) of the samples (including surface and section) was tested using a JSM−7600 F scanning electron microscope (JEOL, Japan) to observe the morphology. An energy dispersive spectrometer (EDS) was employed to analyze the elemental composition. Raman spectroscopy and 2D Raman mapping were performed by a Raman imaging microscope (DXR2xi, Thermo Scientific). Oxygen transmission rate, water vapor transmission rate, and nitrogen transmission rate were measured according to the method in ISO 2556 and GB/T 1037−2021 at 23 °C and 50% RH. The average of three parallel measurements was reported.

## Mechanical properties

The mechanical properties of the samples were measured using a universal testing machine (UTM6530, Shenzhen Suns Technology Stock Co., Ltd.). The samples were cut into a dumbbell shape (25 mm length × 5 mm wide), and the tensile tests were carried out at an extension rate of 5 mm/min at 25 °C. Each sample was measured three times, and then the average values were calculated. Compressive measurements of the cardboard prepared (25 mm length × 25 mm wide × 8 mm high) by paper plastic were also performed using a universal testing machine. The Charpy impact test was performed on a pendulum impact tester for a sample with dimensions of about 150 mm long, 10 mm wide, and 8 mm thick.

## Thermal performance testing

Thermogravimetric analysis (TGA) was characterized by a TG 209 F1 Libra thermobalance (Netzsch, Germany), and the samples were heated from 30 to 800 °C at a rate of 10 °C/min under nitrogen flow. Thermogravimetric infrared (TG−FTIR) analysis of cellulose paper, NIPU, and paper plastic was carried out on a thermogravimetric analyzer (TG209F1, Netzsch, Germany) connected to an FTIR spectrometer (Nicolet IS50, Nicolet, U.S.A.) under airflow conditions. The coefficient of thermal expansion (CTE) of the paper plastic and commercial plastics was measured using a TMA Q400 analyzer (TA Instruments) from −20 to 160 °C.

## Water and solvent stability testing

The paper plastic and cellulose paper samples were cut into rectangular splines and immersed in water to verify the water resistance. The samples were taken out daily, surface moisture was wiped off, and the weight was measured. The morphology before and after a week of immersion was recorded. Water uptake was calculated according to Eq. (1).

$$\text{Water uptake}(\%) = \frac{W - W_0}{W_0} \times 100\% \qquad (1)$$

where $W$ and $W_0$ are the weights of the soaked and original samples, respectively. Simultaneously, the water contact angle measurement of the samples was carried out using a contact angle apparatus (Attention Theta Lite, Bolin Scientific, Sweden) to further explore the hydrophobic properties. About 5 μL of water droplets were placed on the sample surface, and the drop angles of the left and right sides were collected to calculate the average angle. Each sample was tested at least three times and the average value was calculated.

In addition, paper plastics were immersed in different organic solvents to explore solvent resistance. Briefly, the samples were cut into thin strips and soaked in MeOH, EtOH, IPA, acetone, EtOAc, THF, DMF, hexane, and toluene for 7 days, respectively. Then the morphologies of the paper plastics in the sample bottles were observed and recorded. Moreover, the size and morphology of samples before and after immersion in DMF, toluene, DCM, and THF for 7 d were compared.

## Biodegradability test

For the biodegradability test, the samples with a diameter of 7 mm were buried in natural soil (Nanjing, China) at a depth of 10 mm. Samples were taken out for observation periodically, and the morphologies were recorded with a digital camera. The biodegradability of paper plastics was further analyzed by the method GB/T 19277.1−2011, and the amount of $CO_2$ release in the process was recorded. Three groups of 50 g paper plastics were placed in compost containers and kept at 58 ± 2 °C for parallel biodegradation experiments. The thin layer chromatography grade (TLC) cellulose was used as the reference material, and treated in the same method as a contrast. Paper plastics and 11 common plastics (PS, PE, PP, PMMA, PVC, PC, PET, PA6, PA66, PLA, and PBAT) were pyrolyzed by Pyrolysis Gas Chromatography-Mass Spectrometer (Py-GCMS, GC-2030, GCMS-QP2020NX, Shimadzu) at 550 °C, and the types of microplastics that may be produced after degradation of paper plastics were analyzed by comparing gas chromatography and mass spectrometry data.

## Chemical and physical degradation

The sample was cut into small pieces and then hot pressed at 140 °C for 8 h with a 5 MPa force to experience the physical degradation. For chemical degradation, the sample was boiled in a 0.5 M NaOH solution for 3 h. Then the cellulose paper fibers and polymer solution were obtained by filtration and separation. The paper fibers can be used to make complete cellulose paper again.

## All-atom molecular dynamics simulations

All-atom molecular dynamics simulations were performed using large-scale atomic/molecular massively parallel simulator (LAMMPS, 29 Sep 2021 - Update) computational package[56], in which the ReaxFF reactive force field was adopted[57] because of its advantage for simulating complicated reactions without any preconditioning[58].

## Coarse-grained molecular dynamics simulations

The coarse-grained molecular dynamics simulations of cellulose paper and paper plastic were used to elucidate the microstructural evolution and performance improvement. These simulations were performed using LAMMPS[59,60]. The cellulose paper and paper plastic were composed of cellulose microfibrils and NIPU microfibrils. The computational cell sizes utilized in the coarse-grained simulations were 600 nm × 600 nm × 9 nm, with paper microfibrils (cellulose) and NIPU microfibrils randomly distributed in the initial structures. The structural optimizations were subsequently performed in the NPT ensemble at 300 K and a pressure of 1 bar, during which the time step was 25 fs. The densities of cellulose paper (0.71 g/cm³) and paper plastic (1.21 g/cm³) are consistent with corresponding experimental values of 0.67 and 1.20 g/cm³, respectively (Supplementary Fig. 17a). In these simulations, the cellulose microfibrils and NIPU microfibrils were modeled using coarse-grained bead-springs. To match the models with experiments, the diameter of 3 nm used for cellulose microfibrils[3] was adopted in this work. Similarly, the diameter of NIPU microfibrils was also 3 nm. Hence, the equilibrium coarse-grained bond length ($d_0$) of 3 nm was used. The coarse-grained bond potential was given by

$$E_d = \frac{k_d}{2}(d - d_0)^2 \qquad (2)$$

where $d$ is the distance between the coarse-grained beads and $k_d$ is the tensile spring constant. For cellulose microfibrils, the spring constant

$k_d = YA/d_0 = 115\ k_B T/\text{Å}^2$ was obtained from the experimentally measured Young's modulus and the published value[60,61], and the corresponding cross-sectional area $A = \pi d_0^2/4 = 7.07\ \text{nm}^2$. The spring constant $k_d$ of NIPU microfibrils was set as $86\ k_B T/\text{Å}^2$. Considering the slippage failure of microfibrils, the bond-breaking strain of these coarse-grained bonds was about 60% as evident from all-atom molecular dynamics simulations of the failure of microfibrils. The bending behaviors of cellulose microfibrils and NIPU microfibrils were described by

$$E_\theta = \frac{k_\theta}{r_0}\left[1 - cos(\theta - \theta_0)\right] = \frac{l_p k_B T}{r_0}\left[1 - cos(\theta - \theta_0)\right], (0 \le \theta \le 2\pi) \quad (3)$$

where $l_p$ is the persistence length, $r_0$ is the equilibrium distance of microfibrils (3 nm), and $k_B$ and $T$ are the Boltzmann constant and temperature, respectively. The persistence length of cellulose microfibrils (1 μm) and NIPU (1.5 μm) were obtained from the experimental data for cellulose microfibrils[62] and all-atom molecular dynamics simulations for NIPU. The interchain interactions of cellulose microfibrils/cellulose microfibrils, NIPU microfibrils/NIPU microfibrils, and cellulose microfibrils/NIPU microfibrils were described by the Morse-form potentials:

$$E_{pair} = D_0\left(e^{-2\alpha(r-r_0)} - 2e^{-\alpha(r-r_0)}\right), (r \le r_c) \quad (4)$$

where $D_0$, $\alpha$, $r_0$, and $r_c$ represent the binding energy, the inverse length scale for reaching the maximal stretching, the equilibrium distance of microfibrils, and the cut-off distance, respectively. Considering the features of different interchain interactions, the cut-off distances are 2.5 and 1.1 times of the equilibrium distance for the long-range relatively weak interchain interactions (cellulose microfibrils/cellulose microfibrils, and NIPU microfibrils/ NIPU microfibrils) and short-range strong covalent interchain interactions (cellulose microfibrils/NIPU microfibrils), respectively.

### Biocompatibility experiments
The biocompatibility experiments were carried out in Jiangsu Science Standard Medical Testing Co., Ltd., including cytotoxicity, skin irritation, and skin sensitization, which also approved by this company's Animal Welfare Ethics Committee. The cell used in this work is L-929 cells (NCTC clone 929: CCL 1, American Type Culture Collection [ATCC]). The white guinea pigs (male, healthy, not previously used in other experimental procedures, provided by Suzhou Hi-tech Zone Zhenhu Laboratory Animal Technology Co., Ltd [Permit Code: SCXK (SU) 2020-0007]) and New Zealand white rabbits (female, healthy, young adult, nulliparous and not pregnant, provided by Danyang Changyi experimental animal breeding Co., Ltd [Permit Code: SCXK (SU) 2021-0002]) used in the experiment met the requirements of experimental ethics and obtained the corresponding consent forms for ethical review of experimental animals (IACUC number: IACUC22−0045 and IACUC22−0046). Detailed experimental animal information can be found in Supplementary Table 9.

### Life-cycle assessment (LCA)
The aim of this study is to determine the environmental impacts associated with the production of paper plastic and its competitive benchmark plastic films and composite materials using life cycle assessment (LCA)[63,64]. By comparing the environmental impacts at the production stages, this study aims to identify key determinants and corresponding optimization measures. The functional unit (FU) was designed to produce plastic materials with the same tensile strength, normalizing the environmental impact results by tensile strength and density. To enable future comparison with other related materials, the impacts are also reported per kilogram of material. This normalization allows for a comparison of different plastic

materials in terms of both environmental impacts and material properties (Supplementary Note 3). A cradle-to-grave system boundary has been set to consider the impacts originating from raw material preparation to the end-of-life management of the paper plastic. The modeling includes: CSBO preparation, APAC preparation, paper plastic preparation (Supplementary Table 10), composting under aerobic conditions and non-biogenic carbon emissions during composting. The $CO_2$ needed for polymerization can originate from a variety of sources, including carbon sequestration. However, to avoid modeling an overly optimistic scenario, the $CO_2$ has been modeled according to the one used in the chemical industry as "carbon dioxide, in the chemical industry".

The inventory for paper plastic is based on primary data originating from our own laboratory preparation processes. The background data is sourced from the latest available ecoinvent database (ecoinvent v3.11 Cut-Off Unit Processes, released in December 2024). Due to data availability in the life cycle inventory, refined soybean oil and ascorbic acid were used as substitutes for epoxidized soybean oil and L−ascorbic acid, respectively. The life cycle impact assessment (LCIA) was modeled using the latest available OpenLCA version (2.4.0, released on January 2025). The assessment model chosen was Environmental Footprint 3.1 in light of the European Commission's recommendations on measuring and communicating the life cycle environmental performance of products and organizations. This approach also adheres to the operational guidelines of the ISO 14040/14044 standards.

### Techno-economic analysis (TEA)
The technical economics of producing 1 ton of paper plastic in a Chinese factory were evaluated[63]. Material and energy demand data are derived from the raw data of the equipment, while the efficiency of the equipment is assumed to be 90% taking into account the time required for maintenance. In addition, TEA on biobased materials uses 90−95% values. The cost of purchasing equipment and leasing the factory, equipment maintenance and insurance costs, electricity, and labor costs are all taken into account.

### Statistics and reproducibility
All experiments were repeated independently with similar results at least three times.

### Reporting summary
Further information on research design is available in the Nature Portfolio Reporting Summary linked to this article.

## Data availability
The data that support the findings of this study are available within this paper or included in the Supplementary Information, and from the corresponding authors upon request. Source data are provided with this paper.

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

## Acknowledgements

This research was undertaken, in part, thanks to funding from the National Natural Science Foundation of China (Grant No. 32171722) to X.X., the National Natural Science Foundation of China (Grant No. 32401529) to X.Y., the National Natural Science Foundation of China (Grant No. 32471817) to H.L., the National Natural Science Foundation of China (Grant No. 52202288) to L.Y., the National Natural Science Foundation of China (Grant No. 52273091 and Grant No. 22478307) and the Fundamental Research Funds for the Central Universities (Grant No. 691000003) to C.J.C. The numerical calculations in this work were done on the supercomputing system in the Supercomputing Center of Wuhan University. The authors are grateful for the financial support of the University of the Basque Country (Convocatoria de ayudas a grupos de investigación GIU21/010).

## Author contributions

C.J.C., X.X., and H.L. conceived the concept, processing, and structure details and supervised the work. X.Y. and L.Y. carried out most experiments. B.Z. contributed to experiments and characterizations. F.D., L.Q., and L.C. assisted in completing the photographs of samples in the manuscript. E.L. and C.C. contributed to the life cycle-assessment. Y.W., X.J., and E.G. carried out the computational simulation and analyzed the results. X.Y. and L.Y. analyzed the data and co-wrote the manuscript. C.J.C., X.X., H.L., X.Y., and L.Y. revised the manuscript. All authors commented on the submitted version of the manuscript.

## Competing interests

The authors declare no competing interests.

## Additional information

¹National Key Laboratory for Development and Utilization of Forest Food Resources, Jiangsu Co-Innovation Center of Efficient Processing and Utilization of Forest Resources, College of Chemical Engineering, Nanjing Forestry University, Nanjing, China. ²Hubei Biomass-Resource Chemistry and Environmental Biotechnology Key Laboratory, Hubei Provincial Engineering Research Center of Emerging Functional Coating Materials, School of Resource and

Environmental Sciences, Wuhan University, Wuhan, China. [3]National Key Laboratory for Development and Utilization of Forest Food Resources, Institute of Chemical Industry of Forest Products, Chinese Academy of Forestry, Nanjing, China. [4]Department of Engineering Mechanics, School of Civil Engineering, State Key Laboratory of Water Resources and Hydropower Engineering Science, Wuhan University, Wuhan, China. [5]Life Cycle Thinking Group, Department of Graphic Design and Engineering Projects, University of the Basque Country (UPV/EHU), Bilbao, Spain. [6]State Environmental Protection Key Laboratory of Soil Health and Green Remediation, Huazhong Agricultural University, Wuhan, China. [7]These authors contributed equally: Xinxin Yang, Le Yu.
✉e-mail: enlaigao@whu.edu.cn; xuxu200121@njfu.edu.cn; liuhe.caf@gmail.com; chenchaojili@whu.edu.cn

