## [Transparent Peer Review file · Nature Communications]

Rapidly Making Biodegradable and Recyclable Paper Plastic Based on Microwave Radiation Driven Dynamic Carbamate Chemistry

Corresponding Author: Professor Chaoji Chen

Version 0:

Reviewer comments:

Reviewer #1

(Remarks to the Author)

This article mainly studies a new type of "paper plastic", which is prepared based on dynamic carbamate chemistry driven by microwave radiation and has biodegradable and recyclable properties. The article points out that paper is a highly biodegradable material, so it can still maintain its easily degradable properties after modification. In the study, the paper plastic (NIPU) was made by introducing bio-based raw materials such as cyclic carbonated soybean oil and acrylic acid pimaic acid cyclic carbonate into paper. The biodegradation test of this paper plastic in the natural environment showed that after one month of burial, the paper plastic began to show signs of degradation and finally degraded completely within 180 days, while traditional commercial plastics (such as polyethylene) took decades or even hundreds of years to degrade. In addition, the article also emphasized the recyclability of paper plastics. The presence of dynamic carbamate bonds allows paper plastics to be recycled by chemical methods after undergoing biodegradation. Through hot pressing treatment, fragmented paper plastics can be reprocessed into complete materials. The article also explores the life cycle assessment (LCA) and technical and economic analysis (TEA) of paper plastics, indicating that paper plastics have advantages in terms of environmental impact and economic feasibility. Overall, this study demonstrates paper plastics as a promising alternative to traditional plastic packaging materials with good environmental friendliness and economical performance.

1. Insufficient description of the dynamic urethane chemistry: Although the article mentions the formation and cross-linking mechanism of dynamic urethane, the detailed description of its reaction process is relatively brief, which may make it difficult for readers to fully understand the specific mechanism and importance of this chemical reaction. In particular, the lack of in-depth explanation and schematic diagram of the reorganization of dynamic covalent bonds and the interaction with paper fibers may confuse readers in understanding its application in paper plastics.
2. Lack of detailed data on environmental impact and sustainability assessment: Although the article mentions that paper plastics show low impact in multiple environmental impact categories, there is a lack of specific quantitative data and detailed life cycle assessment (LCA) results. This may affect its credibility in practical applications, especially when compared with traditional plastics (such as PE, PET, etc.). The lack of sufficient data support may cause readers to have doubts about its environmental friendliness and sustainability, especially in terms of the resources required in the production process (such as water, electricity) and the biodegradability of the final product.
3. The paper plastic mentioned in the article has a bio-based carbon content of 82%. What advantages does this have over traditional plastics? What is the impact of this high bio-based carbon content on the environment?
4. Results in Supplementary Fig. 33 do not align with common facts that GWPs of HDPE and LDPE are 2.6 and 2.9 kg CO₂e per kg (<https://doi.org/10.1016/j.jclepro.2020.124010>) or published results e.g. ~400 g/cm³.MPa for ABS in Fig 5 of publication <https://doi.org/10.1038/s41893-021-00702-w>. Please check the LCA calculation.
5. What are inputs of CO₂ indicated in Table S8? Are they biogenic carbon sequestered in the feedstock? If so, please specify the terminology as carbon sequestered, similarly in Figure S34.
6. The life cycle assessment (LCA) results mentioned in the article show the greenhouse gas emissions of paper plastics compared with a variety of commercial plastics. What implications do these results have for the selection and application of future materials?

Reviewer #2

(Remarks to the Author)

This manuscript presents an innovative approach to creating a “paper plastic” with enhanced mechanical properties and water resistance while maintaining biodegradability. This study addresses a significant environmental issue by proposing a potential alternative to conventional plastics. The use of bio-based materials and CO₂ as a starting chemical aligns with carbon neutrality goals.

While the research shows promise, several areas require attention and improvement:

1. To demonstrate the dynamic carbamate chemistry of the thermoset polymer in the paper plastic, the recyclability of the pure plastic (without paper) in Fig. 2a should be shown. The recyclability test in the presence of paper (Fig. 6b) is meaningless due to interference. The authors can only claim the viability of “dynamic carbamate chemistry” by showing data that the mechanical strength is maintained after recycling the pure plastic.
2. The experiments conducted by the authors demonstrate bio-decomposability. Biodegradability requires data showing the amount of carbon dioxide generated by microorganisms. The possibility of the plastic part becoming microplastic cannot be excluded in the authors' experiments.
3. The polyethylene value in Fig. 1d is significantly underestimated. It is unclear where the basis for this value comes from.
4. According to the references below and numerous other publications, the CO₂ emission factor for PE is ~2.8 kgCO₂/kg, and for PLA, it is ~0.5 kgCO₂/kg. Supplementary Fig. 33 and related content are erroneous and must be corrected.
(<https://www.climatiq.io/data/emission-factor/d6dd4eb8-0b64-4d04-af59-63cec115f6da>
<https://www.totalenergies-corbion.com/news/low-carbon-footprint-of-pla-confirmed-by-peer-reviewed-life-cycle-assessment/>)

Reviewer #3

(Remarks to the Author)

This work reports the preparation of paper plastic based on cellulose and non-isocyanate polyurethane (NIPU) by utilizing microwave radiation induced curing. The curing is claimed fast, delivering a paper plastic bearing dynamic carbamate bonds. The mechanical properties, solvent resistance, oxygen barrier properties, biocompatibility, biodegradation and recycling ability are assessed. The authors are also reporting Life-cycle assessment (LCA) and technico-economic analysis (TAE) of the process/product.

The paper is interesting, very well presented and illustrated, and easy to read. However, the degree of innovation is very limited to justify publication in Nat. Commun. The authors are introducing “paper plastic” but this is nothing else than a cellulose-based thermoset composite prepared by depositing a formulation composed of two cyclic carbonate products and a polyamine (PEI) on a cellulose paper, followed by curing. Such type of cellulose/NIPU composites have already been prepared and reported (by utilizing a more classical thermal curing), and presented also excellent mechanical properties. They were also reprocessed by exploiting the dynamic behavior of the carbamate groups (see <https://doi.org/10.1016/j.compositesa.2024.108311> for flax/NIPU composites). Moreover, although the manuscript is very well written and presented, it is oversold on many aspects by comparing the performance of the material to conventional plastics that are thermoplastics, and not thermosets nor composites. The comparison is clearly not appropriate (see detailed comments below). More appropriate comparisons should be done, with conclusions that will certainly be different to the present ones... Some info are also missing to permit the reader to reproduce the experiments.

Selected comments:

- Fig 2a and related discussion. What is the NH₂ functionality of PEI ? What is the molar mass of PEI ? There are many different PEI and the authors should give these information to allow the reader to reproduce the experiments.
- Fig 2d and e: the color code is not defined
- Page 7. “A molar ratio of CSBO: (APAC + PEI) = 1 : 1 is adopted to theoretically ensure the complete reaction of cyclic carbonate 125 and amino moieties to form carbamate dynamic bonds... ». The molar ratio used is absolutely unclear. If this ratio is used, there will be an excess of cyclic carbonate coming from CSBO – but it will depend on the functionality of PEI that is not mentioned in the document. Do the authors discuss about molar ratios of the molecules or the functional groups ? – this is not the same. For clarity to the reader, they should express the molar ratios of the functional groups, thus of the cyclic carbonate and primary amine groups.
- Page 9. “we reasoned that the reformation of carbamate moiety between cellulose and NIPU is responsible for the significant interfacial reinforcement and therefore the mechanical properties. » There is no convincing data in the manuscript proving the formation of covalent bonds between the cellulose and the NIPU by transurethanization. Hydrogen bonding between NIPU and the cellulose might also account for the interfacial and mechanical properties reinforcement as it has been shown in <https://doi.org/10.1016/j.compositesa.2024.108311>. This should be discussed in regards to this seminal work.
- Page 12. Comparison of the CTE values and softening/shrinkage of paper plastic when placed at 150 °C with those of commercial plastics (PP, PE, PS, PLA). This comparison is not appropriate as the authors compare thermoplastics (PP, PE, PS, PLA) to a thermoset composite (the paper plastic). This is well-known that thermosets have improved thermal stability compared to thermoplastics ! A fair comparison would be to compare the paper plastic with a similar paper plastic made from another thermoset, for instance an epoxy resin or even, a conventional PU thermoset. The message will be totally different...
- Page 13. “The surface wettability test revealed that the paper plastic demonstrates exceptional hydrophobicity, as

evidenced by a contact angle of over 100° (Fig. 4c), in contrast to that water droplets on the surface of the paper are rapidly absorbed, showing a water of ~160%. » Such type of phrasing oversells the technology. It is evident that coating the hydrophilic cellulosic-based paper by a hydrophobic NIPU will strongly increase the contact angle to the value of NIPU (100.13°, suppl Fig 23). Nothing exceptional here. Moreover, did the authors follow the evolution of the contact angle with time? NIPUs absorb water due to the pending OH groups and the contact angle generally decreases over time.

- Pages 13-14. "The state of paper plastics was found to be less than 20%, leading to a high wet-state tensile strength of 85 MPa (about 72% of dry paper plastic), still surpassing some commercial plastics' tensile strength in dry environments ». Comparison is here not appropriate too. Prepare a similar paper plastic with another thermoset instead of thermoplastics, the message will be different.

- Page 14. "These results indicated that the incorporation of NIPU into the paper network effectively enhances the mechanical properties of paper plastic in wet environments. » This is evident that an hydrophobic coating of NIPU on the hydrophilic paper will enhance its stability in water, and thus its mechanical properties. Nothing new and surprising.

- Page 14. Hot pressing of the paper plastic to form the bag. Exploiting the dynamic behavior of the carbamate bonds to reprocess NIPU thermosets as well as NIPU composites is not new. If the reviewer reads well, there is no experimental section dealing with the procedure used for producing this bag. What is the pressure applied, the contact time and, importantly, the temperature? Generally, NIPUs require a quite high temperature to be reprocessed (160 °C) that induces side reactions such as urea formation that limits further reprocessing steps. Did the authors observe some urea formation (IR is a simple tool to monitor them) and if no, why?

- Fig 4e. As the paper plastic is a composite thermoset, it is normal that the product does not dissolve in organic solvents and presents solvent resistance. This is one of the numerous advantages of thermosets.

- Fig 5d. Once again the comparison of the mechanical performance of the paper plastic with those of commercial plastics, that are thermoplastics is not appropriate. Compare thermoset composites with thermoset composites – with thermosets that are known to interact well with the cellulose (epoxy resin, polyurethane, etc). Compare for instance with a previous work dealing with cellulose-based material/NIPU composites (<https://doi.org/10.1016/j.compositesa.2024.108311>).

- Fig 5e. Comparison for oxygen barrier properties should be done with the best polymers used for that application, thus ethylene vinyl alcohol copolymers (EVOH) and polyvinyl alcohol (PVOH).

Experimental section.

- The mol contents of the different products should be specified, as well as the yields of the products.

- How are recovered the cyclic carbonates after synthesis. Are they degassed to remove dissolved CO₂? Is the catalytic system (TBAL/ascorbic acid) removed, and if yes, how?

- Which equipment is used to coat the monomer mixture on the paper, and how is realized the coating?

- Curing the formulation by microwave radiation-induced curing. How long is the curing step? What is the equipment used and its power? What is the temperature reached by this curing? This latter is important as it will dictate the rate of the reaction but also the content of side products (such as urea bonds).

- Synthesis of APAC. I am very surprised by the reaction conditions that are quite harsh, especially for the first step, the reaction of APA with epichlorhydrine. Is there no ring-opening of the epoxide? Due to the complexity of the final cyclic carbonate (APAC), the provided 1H NMR spectrum is not enough to confirm the structure of the molecule. 13C NMR and 2D NMR should be provided to confirm the structure.

- For the carbonatation, the volume of the reactor should be specified. Are the authors working under constant CO₂ pressure?

- Production of the plastic paper. APAC, CSBO and PEI are mixed at room temperature for 30 min before coating the paper. This is quite surprising to mix for a so long period of time when we know that PEI reacts rapidly at room temperature with external cyclic carbonates, thus those of APAC for instance. It is less reactive with the internal cyclic carbonates of CSBO. Is the viscosity of the mixture increased during mixing? Did the authors check the temperature of their mixture over time as the aminolysis of cyclic carbonates is exothermic? It might be dangerous to prepare such mixture on large volumes. Generally the polyamine is mixed rapidly to the cyclic carbonate and then used directly.

Supplementary information

Suppl. Fig 8 is not 1H NMR spectra. It represents 13C NMR spectra.

Life-cycle assessment (LCA) and technico-economic analysis (TAE). I am not the specialist of these analyses but some of the reported data for the plastic paper seem too good to my opinion. From all LCA/TAE I have read, this is difficult to be better or even close to PE for LCA and TAE of new polymers, especially NIPUs. When I read the protocol for synthesizing APAC from APA, the process is far to be green (large excess of epichlorhydrine, high temperature, etc for the first step), the environmental impact and cost might not be so low. From the reported TAE, paper plastic would be roughly 30% more expensive than PE, this seems very optimistic to me. There is already one interesting report on LCA and TAE of NIPU (<https://doi.org/10.1021/acssuschemeng.4c04046>). Although the cyclic carbonates and polyamines are different, the price of the NIPU is already of more than 5\$/kg. This work should be cited and comparison of LCA/TAE should be done. All these LCA/TAE should be checked by a specialist.

Reviewer comments:

Reviewer #1

(Remarks to the Author)

[Note from the Editor: this reviewer also looked over the responses to reviewer 3, and felt that these comments were also addressed].

The authors have addressed all of my comments and supplemented the paper with additional experimental results in order to address the concerns of the other reviewers.

Reviewer #2

(Remarks to the Author)

The authors have adequately addressed all my comments and concerns. I support the publication of this manuscript in its current form, although I recommend ensuring consistency in terminology such as 'Figure' vs. 'Fig.' and units like 'hours' vs. 'h' or 'days' vs. 'd'.

Itemized list of response to the reviewers' remarks
(Black: Reviewers' remarks; Blue type: Our response)

Reviewer #1:

Comments:

This article mainly studies a new type of "paper plastic", which is prepared based on dynamic carbamate chemistry driven by microwave radiation and has biodegradable and recyclable properties. The article points out that paper is a highly biodegradable material, so it can still maintain its easily degradable properties after modification. In the study, the paper plastic (NIPU) was made by introducing bio-based raw materials such as cyclic carbonated soybean oil and acrylic acid pimaic acid cyclic carbonate into paper. The biodegradation test of this paper plastic in the natural environment showed that after one month of burial, the paper plastic began to show signs of degradation and finally degraded completely within 180 days, while traditional commercial plastics (such as polyethylene) took decades or even hundreds of years to degrade. In addition, the article also emphasized the recyclability of paper plastics. The presence of dynamic carbamate bonds allows paper plastics to be recycled by chemical methods after undergoing biodegradation. Through hot pressing treatment, fragmented paper plastics can be reprocessed into complete materials. The article also explores the life cycle assessment (LCA) and technical and economic analysis (TEA) of paper plastics, indicating that paper plastics have advantages in terms of environmental impact and economic feasibility. Overall, this study demonstrates paper plastics as a promising alternative to traditional plastic packaging materials with good environmental friendliness and economical performance.

Reply to the Reviewer: We thank the Reviewer #1 for the constructive comments, especially pointing out the promising alternative to traditional plastic packaging materials with good environmental friendliness and economic performance of our work.

1. Insufficient description of the dynamic urethane chemistry: Although the article mentions the formation and cross-linking mechanism of dynamic urethane, the detailed description of its reaction process is relatively brief, which may make it difficult for readers to fully understand the specific mechanism and importance of this chemical reaction. In particular, the lack of in-depth explanation and schematic diagram of the reorganization of dynamic covalent bonds and the interaction with paper fibers may confuse readers in understanding its application in paper plastics.

Reply to the Reviewer: We thank the Reviewer #1 for the valuable suggestions. To describe the reaction process in more detail, we have added a schematic diagram of the bond exchange mechanism of the carbamate bond and the interaction of the carbamate bond of the NIPU with the hydroxyl group in cellulose paper. The recombination process of carbamate bonds can be described in the literature¹⁻

³, as shown in the Supplementary Fig. S1a. Meanwhile, we have also supplemented the schematic diagram of the exchange of NIPU network with cellulose paper based on the mechanism of carbamate bond exchange (Supplementary Fig. S1b).

Supplementary Fig. 1 | Schematic diagram of the dynamic carbamate chemistry. a, the bond exchange mechanism of the carbamate bond. b, the schematic diagram of the exchange of NIPU network with cellulose paper.

Updates to the revised manuscript: To enrich the description of the reorganization of dynamic covalent bonds between NIPU and cellulose paper, we have added more careful description in the revised manuscript on Page 5.

“Chemically, both CSBO and APAC contain cyclic carbonate moieties, they can react with the amino groups of PEI to generate the non-isocyanate polyurethane (NIPU) with dynamic carbamate via a catalysis-free mechanism. The literature¹⁻³ confirms that the presence of hydroxyl groups is more conducive to the dynamic bond exchange reaction (transcarbamoylation) of carbamate bonds (Supplementary Fig. 1a) in NIPU. Cellulose is known to be rich in hydroxyl functional groups. Therefore, the dynamic carbamate bond is capable of effectively undergoing bond exchange reactions with hydroxyl groups in cellulose paper (Supplementary Fig. 1b), making it an ideal choice for the preparation of paper plastics.”

References:

1. Lucherelli MA, Duval A, Avérous L. Biobased vitrimers: Towards sustainable and adaptable performing polymer materials. *Progress in Polymer Science* **127**, 101515 (2022).
2. Gomez-Lopez A, Elizalde F, Calvo I, Sardon H. Trends in non-isocyanate polyurethane (NIPU) development. *Chem Commun* **57**, 12254-12265 (2021).
3. Fortman DJ, Brutman JP, Cramer CJ, Hillmyer MA, Dichtel WR. Mechanically activated, catalyst-free polyhydroxyurethane vitrimers. *J Am Chem Soc* **137**, 14019-14022 (2015)

2. *Lack of detailed data on environmental impact and sustainability assessment: Although the article mentions that paper plastics show low impact in multiple environmental impact categories, there is a lack of specific quantitative data and detailed life cycle assessment (LCA) results. This may affect its credibility in practical applications, especially when compared with traditional plastics (such as PE, PET, etc.). The lack of sufficient data support may cause readers to have doubts about its environmental friendliness and sustainability, especially in terms of the resources required in the production process (such as water, electricity) and the biodegradability of the final product.*

Reply to the Reviewer: We thank the Reviewer #1 for the thoughtful suggestions. We agree that it is important to provide sufficient data to replicate the LCA part so that future researchers can identify environmental hotspots and continue our work. Therefore, we invited Prof. Erlantz Lizundia, an expert in the field of LCA research, to re-calculate the LCA and add Prof. Erlantz Lizundia as a co-author to the list of authors. The information on how the LCA was carried out has been expanded in both the Discussion and Methods sections. We modified Fig. 6d, Supplementary Figs. 39 and 40 based on the results of the LCA recalculation. In addition, we now provide clear and unambiguous information on the database, software, and impact assessment methodology used. The modelling for diglycidyl ester of acrylic acid, for tetrabutylammonium iodide, and polyethyleneimine is described in detail Supplementary Note 3. In addition, the information used to determine the impact of petroleum-based materials is provided, detailing the specific database entry, density, and tensile strength. We believe that the additional detail provided is sufficient to support future comparisons and decision-making.

Updates to the revised manuscript: To enrich the description of the specific quantitative data and detailed life cycle assessment (LCA) results, we have added more careful description in the revised manuscript on Pages 27–29. More clear and unambiguous information on the database, software, and impact assessment methodology used in our work were also supplemented in the Methods part of the revised manuscript on Pages 37–38.

“Life cycle assessment (LCA) was applied to determine the cradle-to-grave environmental impacts of paper plastic; i.e. from resource extraction, to end-of-life (EoL). The climate change potential is used to understand the environmental sustainability of the material because it quantifies the greenhouse gas emissions driving climate change, a major global challenge nowadays. Besides, it also provides a common reference for comparing with other products or processes. The climate change potential (normalized to tensile strength as done by Xia et al.¹) in Fig. 6d shows a significantly lower value for paper plastic over 14 potentially competitive benchmark plastic films and composite materials. Such a remarkable low carbon footprint originates from the manufacturing process simplicity, significant biogenic carbon content, and high strength of paper plastic. As shown in Supplementary Fig. 39, the manufacturing of paper plastic (cradle-to-gate) presents a carbon footprint of 6.81 kg CO₂ equiv kg⁻¹, which is just under average for conventional plastics and composites. As a matter of fact, obtained footprint value is just slightly above the cradle-to-gate carbon footprint of 2.6 and 2.9 kg CO₂ equiv kg⁻¹ of HDPE and LDPE², or the ~4.75 kg CO₂ equiv kg⁻¹ reported by Liang et al.³ for nonisocyanate

polythiourethane production (no biogenic credits), a sustainable alternative to conventional plastics. However, it should be noted that the obtained footprint represents a conservative value as the PEI modelled has a carbon footprint of 21.74 (lower values are more likely for industrially processed PEI). Furthermore, the biodegradability and large biogenic content of the paper plastic are translated into reduced CO₂ emission at the EoL. The incineration of conventional plastics and composite materials is subjected to additional burdens because the additional energy input for burning plastics, as well as for the CO₂ emissions originating from the fossil carbon that remains in the plastics (this adds approximately 6 kg CO₂ equiv kg⁻¹). On the contrary, materials from bio-based feedstock can be composted (which has a lower footprint than incineration) and the CO₂ formed during aerobic composting can be accounted as burden free because it will be included in the short-term carbon cycle again⁴. The footprint per kilogram of final material is therefore lower, which together with its good mechanical properties makes paper plastic a low carbon material when the tensile strength is taken as a functional unit (as shown in Fig. 6d).

As shown in Supplementary Fig. 40, the environmental impact of paper plastic is primarily attributed to PEI, which contributes 11.5% to 54.6% of the impact, ranking first in 13 out of the 16 environmental impact categories. Paper, soybean oil and acrylic acid are the other two major contributors, especially in the categories of land use, marine eutrophication and non-carcinogenic human toxicity, respectively. Unlike other studies, the energy consumption in paper plastic production is very low, with a maximum contribution of 0.1%.” on Pages 27–29.

“Life-cycle assessment (LCA)

The aim of this study is to determine the environmental impacts associated with the production of paper plastic and its competitive benchmark plastic films and composite materials using life cycle assessment (LCA)^{5, 6}. By comparing the environmental impacts at the production stages, this study aims to identify key determinants and corresponding optimization measures. The functional unit (FU) was designed as producing plastic materials with the same tensile strength, normalizing the environmental impact results by tensile strength and density. To enable future comparison with other related materials, the impacts are also reported per kilogram of material. This normalization allows for a comparison of different plastic materials in terms of both environmental impacts and material properties (Supplementary Note 3). A cradle-to-grave system boundary has been set to consider the impacts originating from raw material preparation to the end-of-life management of the paper plastic. The modeling includes: CSBO preparation, APAC preparation, paper plastic preparation (Supplementary Tab. 9), composting under aerobic conditions and non-biogenic carbon emissions during composting. The CO₂ needed for polymerization can originate from a variety of sources, including carbon sequestration. However, to avoid modelling an overly optimistic scenario, the CO₂ has been modeled according to the one used in the chemical industry as “carbon dioxide, in chemical industry”.

The inventory for paper plastic is based on primary data originating from our own laboratory preparation processes. The background data is sourced from the latest available ecoinvent database (ecoinvent v3.11 Cut-Off Unit Processes, released on December 2024). Due to data

availability in the life cycle inventory, refined soybean oil and ascorbic acid were used as substitutes for epoxidized soybean oil and L-ascorbic acid, respectively. The life cycle impact assessment (LCIA) was modeled using the latest available OpenLCA version (2.4.0, released on January 2025). The assessment model chosen was Environmental Footprint 3.1 in the light of the European Commission's recommendations on measuring and communicating the life cycle environmental performance of products and organizations. This approach also adheres to the operational guidelines of the ISO standards.” on Pages 37–38.

Updates to the revised Supplementary Information: According to the Reviewer #1’s comments, we added a more precise description in the revised Supplementary Information.

“Supplementary Note 3: Environmental impacts of the paper plastic compared to benchmark plastic films and composite materials.

The Life Cycle Inventory (LCI) data of the upstream production of chemicals, electricity, and water were collected from the ecoinvent v3.11 database and literature. The LCI data for diglycidyl ester of acrylic acid production was estimated based on the process that uses acrylic acid and epichlorohydrin as feedstock. The electricity consumption for the 4 h reaction at 117 °C is estimated to be 1.72 kWh using an Asynt ADS-HP-NT stir plate with a device power of 616 W (<https://www.asynt.com/wp-content/uploads/2011/07/Asynt-Hotplate-Manual-V1.0.pdf>) at a 70% workload. Filtering (0.1 h) and vacuum drying is needed (1 h vacuum drying) for the final material. The electricity consumption of 4.2 Wh is modeled for filtering using a Buchner filtration with a diaphragm pump having a power of 60 W (https://assets.fishersci.com/TFS-Assets/CCG/EU/Welch-Vacuum-Technology/brochures/12139%20FB%20Vacuum%20pumps_EN.pdf) at a 70% workload. For the drying, the electricity consumption of 1.15 kWh is modeled using an oven with a power of 1400 W equipped with a 248 W pump at a 70% workload.

The LCI for tetrabutylammonium iodide (TBAI) production was estimated based on a commercial process that reacts tributylamine with 1-iodobutane. Briefly, the reaction follows:

The reaction takes place in acetonitrile (1.5 L) at 50 °C for 18 h. The electricity consumption of 7.76 kWh is modeled using an Asynt ADS-HP-NT stir plate with a device power of 616 W (<https://www.asynt.com/wp-content/uploads/2011/07/Asynt-Hotplate-Manual-V1.0.pdf>) at a 70% workload. Hexane (0.5 L), and ethanol (1 L) are added for washing and recrystallization. Filtering (0.1 h) and vacuum drying is needed (12 h vacuum drying) for the final material. The electricity consumption of 4.2 Wh is modeled for filtering using a Buchner filtration with a diaphragm pump having a power of 60 W (https://assets.fishersci.com/TFS-Assets/CCG/EU/Welch-Vacuum-Technology/brochures/12139%20FB%20Vacuum%20pumps_EN.pdf) at a 70% workload. For the drying, the electricity consumption of 13.84 kWh is modeled using an oven with a power of 1400 W equipped with a 248 W pump at a 70% workload. Methyl iodide (CH₃I) is utilized as a proxy material for 1-iodobutane due to lack of information in the database utilized (they are

structurally similar and a similar impact is expected), while iodine (I₂) is used to balance the iodine content. In summary, 1 kg of TBAI is modeled with 630 g of CH₃I and 420 g of I₂ (5 wt% losses considered during the synthesis). Generated waste is treated as hazardous solvent.

The LCI data for polyethyleneimine production was estimated based on the process that uses monoethanolamine, hydrogen chloride, chlorosulfuric acid and sodium hydroxide as feedstock. Ethyleneimine is first formed, followed by its polymerization. Monoethanolamine is reacted with HCl and the resulting product is treated with chlorosulfuric acid (sulfuric acid is used as a proxy material due to lack of data in the database). The polymerization process with NaOH as a base catalyst by ring opening polymerization. A 95 % reaction yield is estimated.

The LCIA of the paper plastic was first conducted for 1000 kg of paper plastic, and then the results were converted based on a second FU that takes the tensile strength into consideration, as shown in Eq. (1).

$$EI_p' = EI_p \times \frac{\rho_p}{\sigma_p} \quad (1)$$

where EI_p' is the environmental impact of the paper plastic using the second FU that was converted based on the density (ρ_p) and the tensile strength of the material (σ_p). EI_p is the environmental impact per 1000 kg of paper plastic. The ratio of density to tensile strength is used in the second FU as it considers the impacts of material properties on the functionality of the materials¹. The density and tensile strength of the paper plastic in this study was measured to be 1.245 g·cm⁻³ and 125 MPa, respectively. Obtained environmental impacts were compared to benchmark plastic films and composite materials according to the ecoinvent v3.11 database and the Environmental Footprint 3.1 assessment method. When the data corresponding to the film was not available on the database, we implemented a combination of plastic production (granulate in most of the cases) with extrusion for a plastic film. The entries selected are:

- packaging film production, low density polyethylene | packaging film, low density polyethylene | Cutoff, U - RoW
- polyurethane production, rigid foam | polyurethane, rigid foam | Cutoff, U - RER
- polyethylene terephthalate production, granulate, bottle grade | polyethylene terephthalate, granulate, bottle grade | Cutoff, U - RoW
- polylactic acid production, granulate | polylactic acid, granulate | Cutoff, U - GLO
- polypropylene production, granulate | polypropylene, granulate | Cutoff, U - RoW
- acrylonitrile-butadiene-styrene copolymer production | acrylonitrile-butadiene-styrene copolymer | Cutoff, U - RoW
- polybutadiene production | polybutadiene | Cutoff, U - RoW
- polycarbonate production | polycarbonate | Cutoff, U - RoW
- polyvinyl chloride production, unspecified polymerisation, weighted average | polyvinyl chloride, unspecified polymerisation, weighted average | Cutoff, U - RoW
- polyvinylfluoride production | polyvinylfluoride | Cutoff, U - RoW
- polyvinylidenechloride production, granulate | polyvinylidenechloride, granulate | Cutoff, U - RoW

- *market for polymethyl methacrylate / polymethyl methacrylate / Cutoff, U - RoW*
- *glass fibre reinforced plastic production, polyamide, injection moulded / glass fibre reinforced plastic, polyamide, injection moulded / Cutoff, U - RoW*
- *carbon fibre reinforced plastic, injection moulded / carbon fibre reinforced plastic, injection moulded / Cutoff, U – GLO*
- *extrusion, plastic film / extrusion, plastic film / Cutoff, U - RoW*

The end-of-life modeling includes the fossil-CO₂ emissions during combustion or biodegradation, together with the following processed have been selected:

- *paper plastic: treatment of biowaste, industrial composting / compost / Cutoff, U - RoW*
- *benchmark plastic films and composite materials: treatment of waste plastic, mixture, municipal incineration / waste plastic, mixture / Cutoff, U - RoW”*

Furthermore, the density and tensile strength values in Supplementary Tab. 10 have been utilized to normalize the environmental impacts of benchmark plastic films and composite materials.

Revised Supplementary Tab. 10 | Density and tensile strength used to normalize the environmental impacts.

Material	Density (g·cm⁻³)	Tensile strength (MPa)
Acrylonitrile butadiene styrene (ABS)	1.100	45.0
Polypropylene (PP)	0.900	31.8
Polyethylene terephthalate (PET)	1.345	60.4
Low-density polyethylene (LDPE)	0.949	32.8
Polybutadiene (PB)	0.910	15.0
Polycarbonate (PC)	1.018	65.0
PU rigid foam (PU)	0.200	2.3
Polylactic acid (PLA)	1.250	55.0
Polyvinyl chloride (PVC)	1.360	37.0
Polyvinylfluoride (PVF)	1.470	50.0
Polyvinylidenechloride (PVDC)	1.680	67.0
Poly(methyl methacrylate) (PMMA)	1.190	75.0
Polyamide/glass fiber (PA/GF)	1.340	91.0
ABS/carbon fiber (ABS/CF)	1.265	106.2

Revised Fig. 6 | Environmental impacts of paper plastic. **a**, Degradation process of paper plastic, and polyethylene (PE) buried in soil (Nanjing). **b**, Physical recyclability of paper plastics. Shredded paper plastic recovered to intact paper plastic after hot pressing and cross-section SEM image of the reprocessed paper plastic. **c**, Chemical recyclability of paper plastics. Photos show the degradation of paper plastic in 0.5 M NaOH solution and the regeneration of paper from recyclables. **d**, *Cradle-to-grave* climate change potential (normalized to tensile strength) of the paper plastic compared to 14 potentially competitive benchmark plastic films and composite materials using life cycle assessment (LCA). **e**, Comparison of the economic cost of paper plastic to commercial plastics (PE, PET, PP, PLA, and PCL) via TEA.

Supplementary Fig. 39 | Comparison of the climate change potential of paper plastic to a variety of benchmark plastic films and composite materials according to life cycle assessment (LCA). Left: climate change per 1 kg of material. Right: climate change normalized to tensile strength.

Supplementary Fig. 40 | Environmental Impact Contribution of Each Unit in Plastic Paper Production.

References:

1. Xia Q, et al. A strong, biodegradable and recyclable lignocellulosic bioplastic. *Nature Sustainability*, 627-635 (2021).

2. Benavides PT, Lee U, Zarè-Mehrjerdi O. Life cycle greenhouse gas emissions and energy use of polylactic acid, bio-derived polyethylene, and fossil-derived polyethylene. *Journal of Cleaner Production* **277**, 124010 (2020).
3. Liang C, *et al.* Techno-economic Analysis and Life Cycle Assessment of Biomass-Derived Polyhydroxyurethane and Nonisocyanate Polythiourethane Production and Reprocessing. *ACS Sustainable Chemistry & Engineering* **12**, 12161-12170 (2024).
4. Garcia R, Freire F. Carbon footprint of particleboard: a comparison between ISO/TS 14067, GHG Protocol, PAS 2050 and Climate Declaration. *Journal of Cleaner Production* **66**, 199-209 (2014).
5. Chen L, *et al.* Biomass waste-assisted micro(nano)plastics capture, utilization, and storage for sustainable water remediation. *The Innovation* **5**, 100655 (2024).
6. Huang J, Wang S, Chen J, Chen C, Lizundia E. Environmental Sustainability of Natural Biopolymer - Based Electrolytes for Lithium Ion Battery Applications. *Advanced Materials*, (2025).

3 The paper plastic mentioned in the article has a bio-based carbon content of 82%. What advantages does this have over traditional plastics? What is the impact of this high bio-based carbon content on the environment?

Reply to the Reviewer: We thank the Reviewer #1 for the thoughtful comments. Bio-based carbon content is an important means of assessing the proportion of renewable resource components in materials, so the bio-based carbon content of paper plastics was measured by ASTM D6866 method in our work, which was 82%. Compared with traditional plastics, bio-based plastics are environmentally friendly, low-carbon and environmentally friendly, with renewable raw materials, which will reduce the dependence of the plastics industry on fossil raw materials and promote the transformation of the plastics industry from polluting to green, sustainable and circular economy.

Furthermore, in recent years, numerous countries have sequentially introduced bio-based carbon certification programs or labels tailored to their respective developmental requirements. The accurate determination of bio-based carbon content is crucial for ensuring that products comply with quality standards, advancing sustainable development, and informing environmental protection policies. Therefore, the high bio-based carbon content helps the product to better comply with the needs of environmental protection policies. Meanwhile, in the context of the world's active implementation of the strategy of "carbon peak and carbon neutrality", the use of bio-based raw materials for the preparation of products can help reduce the carbon footprint in the production process of materials and mitigate global climate change.

4. Results in Supplementary Fig. 33 do not align with common facts that GWPs of HDPE and LDPE are 2.6 and 2.9 kg CO₂e per kg (<https://doi.org/10.1016/j.jclepro.2020.124010>) or published results e.g. ~400 g/cm³.MPa for ABS in Fig 5 of publication <https://doi.org/10.1038/s41893-021-00702-w>. Please check the LCA calculation.

Reply to the Reviewer: We thank Reviewer #1 for the constructive comments. The Reviewer is correct that the calculated carbon footprint values in Supplementary Fig. 39 overestimate the actual footprint values. We have reviewed our calculations using the latest available OpenLCA software (2.4.0, released January 2025), the latest available ecoinvent database (ecoinvent v3.11 Cut-Off Unit Processes, released December 2024) and the Environmental Footprint 3.1 methodology. The new results are reported as Supplementary Fig. 39, and are closer to the values suggested by the Reviewer and online data (e.g. <https://www.recyclingtoday.com/news/recycled-pp-hdpe-lower-carbon-footprint-pet/>). The data provided in Supplementary Fig. 39 consider a *cradle-to-grave* system boundary, where the manufacturing is shown in orange, and *end-of-life* are shown as a function of the footprint from incineration and fossil CO₂ emission. In addition, the suggested references have been included.

Supplementary Fig. 39 | Comparison of the *climate change potential* of paper plastic to a variety of benchmark plastic films and composite materials according to life cycle assessment (LCA). Left: climate change per 1 kg of material. Right: climate change normalized to tensile strength.

5. What are inputs of CO₂ indicated in Table S8? Are they biogenic carbon sequestered in the feedstock? If so, please specify the terminology as carbon sequestered, similarly in Figure S34.

Reply to the Reviewer: We thank the Reviewer #1 for the inspiring suggestions. We apologize for the lack of detail. This CO₂ is required for polymerization of cyclic carbonate, which can originate from a variety of sources, including carbon sequestration. However, to avoid modelling an overly optimistic scenario, we have chosen to model CO₂ used in the chemical industry as “carbon dioxide, in chemical industry” (from the ecoinvent v3.11 database). This information is now included in the methodology part.

Updates to the revised manuscript: According to the Reviewer #1's comments, we added a more precise description in the Methods of the revised manuscript on Page 37.

“The modeling includes: CSBO preparation, APAC preparation, paper plastic preparation (Supplementary Tab. 9), composting under aerobic conditions and non-biogenic carbon emissions during composting. The CO₂ needed for polymerization can originate from a variety of sources, including carbon sequestration. However, to avoid modelling an overly optimistic scenario, the CO₂ has been modeled according to the one used in the chemical industry as “carbon dioxide, in chemical industry”.”

6. *The life cycle assessment (LCA) results mentioned in the article show the greenhouse gas emissions of paper plastics compared with a variety of commercial plastics. What implications do these results have for the selection and application of future materials?*

Reply to the Reviewer: We thank the Reviewer #1 for the constructive comments. Reporting the cradle-to-grave greenhouse gas emissions of paper-plastic provides essential information on the environmental sustainability of the developed material. In our work, we provide the climate change potential, which takes into account all the greenhouse gases generated by the paper-plastic (and provides an equivalent to CO₂), as this is considered to be the most relevant impact metric in the current climate change scenario. Providing the greenhouse gas emissions of the paper-plastic serves as a quick but accurate metric to understand whether the developed material is environmentally sustainable. In addition, reporting the carbon footprint assessment is in line with regulatory requirements, corporate sustainability goals and consumer awareness. In our work, we use the greenhouse gas emissions metric to highlight the environmentally sustainable nature of the developed material, which also benefits from its biogenic carbon content (e.g., the benefits of biogenic carbon are not reflected in other impact categories). Finally, we also provide the distribution of greenhouse gas emissions per life cycle stage (and material type), which helps to identify areas for future improvement.

Reviewer #2:

Comments:

This manuscript presents an innovative approach to creating a “paper plastic” with enhanced mechanical properties and water resistance while maintaining biodegradability. This study addresses a significant environmental issue by proposing a potential alternative to conventional plastics. The use of bio-based materials and CO₂ as a starting chemical aligns with carbon neutrality goals. While the research shows promise, several areas require attention and improvement:

Reply to the Reviewer: We thank Reviewer #2 for the positive comments to our work, particularly for highlighting its potential as an alternative to conventional plastics in addressing environmental concerns.

1. To demonstrate the dynamic carbamate chemistry of the thermoset polymer in the paper plastic, the recyclability of the pure plastic (without paper) in Fig. 2a should be shown. The recyclability test in the presence of paper (Fig. 6b) is meaningless due to interference. The authors can only claim the viability of “dynamic carbamate chemistry” by showing data that the mechanical strength is maintained after recycling the pure plastic.

Reply to the Reviewer: We genuinely thank the Reviewer #2 for the valuable comments. We agree with the Reviewer that recovery experiments of the pure polymer in the paper plastic (non-isocyanate polyurethane (NIPU)) need to be provided to demonstrate dynamic carbamate chemistry. Therefore, two recovery methods (physical hot pressing and chemical degradation) are used to explore the dynamic properties of carbamate containing β -hydroxyl groups.

Recyclability Test of Pure Polymer NIPU via Physical Hot Pressing. The carbamate bond containing β -OH in the NIPU structure undergoes a dynamic transcarbamoylation reaction after being stimulated by the external temperature, which in turn rearranges the network structure (Supplementary Fig. 1). Therefore, we put the shredded NIPU under a hot press machine and carry out the hot pressing reprocessing process at a temperature and pressure of 120 °C and 5 MPa. Since the carbamate bond in the structure is bonded with the hydroxyl group, which promotes the network regeneration at the fracture of the material, the material can be effectively cross-linked after one hour of hot pressing (Supplementary Fig. 38a). Meanwhile, the mechanical properties of the reprocessed NIPU are tested, which can be restored to more than 90% of the original strength after 2 hours of hot pressing. The scratches on the surface of NIPU can also be self-healing after heat treatment.

Supplementary Fig. 38 | Recyclability Test of Pure Polymer NIPU. a, Physical Hot Pressing. b, The stress-strain curves of original NIPU and recycled NIPUs. c, The recovery rates of recycled NIPUs with different methods. d, Self-healing experiments of surface scratches. e, Chemical degradation.

Recyclability Test of Pure Polymer NIPU via chemical degradation. In addition, the small molecule alcohol containing hydroxyl groups (ethanol) is used as a solvent to promote the depolymerization of the NIPU polymer network after breaking through the dynamic exchange of hydroxyl groups and carbamate bonds, so that NIPU can be degraded. After collecting the degraded mixed solution system and rotary evaporation to remove the excess alcohol, the mixture was poured into the Teflon mold and solidified again at 120 °C to prepare NIPU (Supplementary Fig. 38e). The tensile strength of NIPU obtained after re-curing can be restored to 84% of the initial value, which also confirms the dynamics of the network structure. In addition, the recovery rate of chemical degradation is slightly lower than that of the recycled sample after physical hot pressing, which may be due to the fact that some of the small molecule ethanol is still attached to the polymer network, resulting in a slightly reduced degree of cross-linking and a slightly lower recovery rate of mechanical properties than physical recovery.

Updates to the revised Supplementary Information: According to the Reviewer #2's comments, we added a more careful description in the revised manuscript and Supplementary Information.

“Moreover, the recovery experiments of the pure polymer in the paper plastic (NIPU) were provided to demonstrate dynamic carbamate chemistry more clearly (Supplementary Fig. 38). The re-curing NIPUs obtained by physical hot pressing and chemical degradation confirm the dynamic properties of the network structure.” in the revised manuscript on Pages 23.

“The recovery experiments of the pure polymer in the paper plastic (NIPU) were provided to demonstrate dynamic carbamate chemistry more clearly. The physical hot pressing and chemical degradation are used to explore the dynamic properties. The carbamate bond containing β -OH in the NIPU structure undergoes a dynamic transcarbamoylation reaction after being stimulated by the external temperature, which in turn rearranges the network structure (Supplementary Fig. 1). Therefore, we put the shredded NIPU under a hot press machine and carry out the hot pressing reprocessing process at a temperature and pressure of 120 °C and 5 MPa. Since the carbamate bond in the structure is bonded with the hydroxyl group, which promotes the network regeneration at the fracture of the material, the material can be effectively cross-linked after 2 hours of hot pressing (Supplementary Fig. 38a). Meanwhile, the mechanical properties of the reprocessed NIPU are tested, which can be restored to more than 90% (Supplementary Fig. 38b and 38c) of the original strength after hot pressing. The scratches on the surface of NIPU can also be self-healing after heat treatment. In addition, the ethanol is used as a solvent to promote the depolymerization of the NIPU polymer network after breaking through the dynamic exchange of hydroxyl groups and carbamate bonds, so that NIPU can be chemical degraded. After collecting the degraded mixed solution system, after rotary evaporation to remove the excess alcohol, the mixture was poured into the Teflon mold and solidified again at 120 °C to prepare NIPU (Supplementary Fig. 38e). The tensile strength of NIPU obtained after re-curing can be restored to 84% (Supplementary Fig. 38b and 38c) of the initial value, which also confirms the dynamics of the network structure. In addition, the recovery rate of chemical degradation is slightly lower than that of the recycled sample after physical hot pressing, which may be due to the fact that some of the small molecule ethanol is still attached to the polymer network, resulting in a slightly reduced degree of cross-linking and a slightly lower recovery rate of mechanical properties than physical recovery.” in the revised Supplementary Information.

2. The experiments conducted by the authors demonstrate bio-decomposability. Biodegradability requires data showing the amount of carbon dioxide generated by microorganisms. The possibility of the plastic part becoming microplastic cannot be excluded in the authors' experiments.

Reply to the Reviewer: We thank the Reviewer #2 for the thoughtful suggestions. More experiments (material composting degradation and microplastics testing) have been added to explore the biodegradability of paper plastics.

The amount of carbon dioxide generated by microorganisms during the degradation process.

The biodegradability of paper plastics was further analyzed by the method GB/T 19277.1-2011, and the amount of CO₂ release in the process was recorded. Due to time limitations, the test was conducted

for a duration of 45 days. In subsequent work, we plan to conduct more comprehensive biodegradation experiments. Three groups of 50 g paper plastics were placed in compost containers and kept at 58 ± 2 °C for parallel biodegradation experiments. The thin layer chromatography grade (TLC) cellulose was used as the reference material, and treated in the same method as a contrast. The amount of CO₂ emitted gradually increases over time as paper plastics are degraded (Supplementary Fig. 36). Based on the cumulative amount of CO₂ released, the biodegradation percentage of paper plastic was $45.1\% \pm 5.5\%$ at 45 days, and the biodegradation percentage of the reference material was $74.4\% \pm 1.0\%$ ($> 70\%$), indicating that the experimental results were reliable and valid. The relative biodegradation rate of paper plastic compared with the reference material is 61%, which also indicates that paper plastic has obvious biodegradability.

Supplementary Fig. 36 | Determination of the ultimate aerobic biodegradability of paper plastics under controlled composting conditions. a, Physical diagram of paper plastic compost before and after degradation. b, CO₂ emission curve during degradation. c, Biodegradability curve during degradation.

Exploring the possibility of paper plastic becoming microplastic.

In addition, we also investigated the possibility of microplastics being produced after the degradation of paper plastics. Paper plastics and 11 common plastics (PS, PE, PP, PMMA, PVC, PC, PET, PA6, PA66, PLA, and PBAT) were pyrolyzed by Pyrolysis Gas Chromatography-Mass Spectrometer (Py-GCMS) at 550 °C, and the types of microplastics that may be produced after degradation of paper plastics were analyzed by comparing gas chromatography and mass spectrometry data. The characteristic fragments, quantitative ions, standard curves, R², and limits of quantification of microplastics produced by 11 standard plastic products are summarized in the Supplementary Tab. 11. From the comparison of gas chromatography and mass spectrometry data, it can be seen that 11 common microplastics were not detected in paper plastics (Supplementary Tab. 12). Therefore, to a

certain extent, it can be considered that paper plastics are biodegraded during the degradation process, and there is no obvious production of microplastics (Supplementary Fig. 37).

Supplementary Tab. 11 | Summary of microplastic characteristic fragments, quantitative ions, standard curves, R² and limits of quantification produced by 11 standard plastic products.

Plastic	Characteristic fragment	Quantitative ions (m/z)	Standard curves	R ²	Limits of quantification (LOQ, µg)
PS	Styrene trimer	91	Y=11950370X-247312.1	0.9998	0.02
PE	1-Decene	111	Y=700254.5X-169232.9	0.9992	0.22
PP	2,4-Dimethyl-1-heptene	70	Y=990671.3X-5387.353	0.9966	0.02
PMMA	Methyl methacrylate	100	Y=21752830X-974976.3	0.9973	0.02
PVC	Naphthalene	128	Y=1677993X-399449.8	0.9995	0.06
PC	Bisphenol A	228	Y=2836385X-237842.2	0.9994	0.02
PET	Vinyl benzoate	105	Y=766762.9X-171277.3	0.9970	0.10
PA6	Caprolactam	85	Y=6070242X-1058689	0.9967	0.06
PA66	Cyclopentanone	84	Y=4642579X-724033.6	0.9985	0.06
PLA	DL-Lactide	56	Y=2031387X-630451.4	0.9988	0.10
PBAT	(S,S)-2,3-Butanediol	72	Y=4262826X-295504.4	0.9966	0.10

Supplementary Tab. 12 | Summary of microplastics test results for paper plastics.

Name	Microplastic content in the sample (µg/g)										
	PS	PE	PP	PMMA	PVC	PC	PET	PA6	PA66	PLA	PBAT
Paper plastic	N.D.	N.D.	N.D.	N.D.	N.D.	N.D.	N.D.	N.D.	N.D.	N.D.	N.D.
N.D. indicates below the detection limit or not detected.											

Supplementary Fig. 37 | Exploring whether paper plastics produce microplastics. Mass spectra of the pyrolyzed paper plastic compared to the standard plastics.

Updates to the revised manuscript: According to the Reviewer #2's comments, we added a more careful description in the Methods (Biodegradability test), Results (Environmental impacts of paper plastic) and Supplementary Information.

“The biodegradability of paper plastics was further analyzed by the method GB/T 19277.1-2011, and the amount of CO₂ release in the process was recorded. Three groups of 50 g paper plastics were placed in compost containers and kept at 58 ± 2 °C for parallel biodegradation experiments. The thin layer chromatography grade (TLC) cellulose was used as the reference material, and treated in the same method as a contrast. Paper plastics and 11 common plastics (PS, PE, PP, PMMA, PVC, PC, PET, PA6, PA66, PLA, and PBAT) were pyrolyzed by Pyrolysis

Gas Chromatography-Mass Spectrometer (Py-GCMS) at 550 °C, and the types of microplastics that may be produced after degradation of paper plastics were analyzed by comparing gas chromatography and mass spectrometry data.” on Page 33 of revised manuscript.

“In addition, data on the amount of CO₂ produced by microorganisms during the biodegradability process (Supplementary Fig. 36) and the possibility of microplastics being produced (Supplementary Fig. 37) after the degradation of paper plastics are measured. The results confirm the paper plastics are biodegraded during the degradation process, and there is no obvious production of microplastics.” on Page 22 of revised manuscript.

“The amount of CO₂ emitted gradually increases over time as paper plastics are degraded (Supplementary Fig. 36). Based on the cumulative amount of CO₂ released, the biodegradation percentage of paper plastic was 45% ± 5.5% at 45 days, and the biodegradation percentage of the reference material was 74.4% ± 1.0% (> 70%), indicating that the experimental results were reliable and valid. The relative biodegradation rate of paper plastic compared with the reference material is 61%, which also indicates that paper plastic has obvious biodegradability.” in the revised Supplementary Information.

“The characteristic fragments, quantitative ions, standard curves, R², and limits of quantification of microplastics produced by 11 standard plastic products are summarized in the Supplementary Tab. 11. From the comparison of gas chromatography and mass spectrometry data, it can be seen that 11 common microplastics were not detected in paper plastics (Supplementary Tab. 12). Therefore, to a certain extent, it can be considered that paper plastics are biodegraded during the degradation process, and there is no obvious production of microplastics (Supplementary Fig. 37).” in the revised Supplementary Information.

3. The polyethylene value in Fig. 1d is significantly underestimated. It is unclear where the basis for this value comes from.

Reply to the Reviewer: We thank the Reviewer #2 for the thoughtful suggestions. The reviewer was correct, and Fig. 1d did have some mistakes when drawing. We have made corrections to Fig. 1d and Supplementary Fig. 7 based on data from some references¹⁻⁴ and supplemented the related references in the revised manuscript. The price of polyethylene should be more expensive than paper, but slightly lower than that of paper plastic. Although polyethylene is a traditional thermoplastic, its structural stability is high, it is difficult to degrade and recycle, and incineration or landfill disposal of waste polyethylene plastic will cause environmental pollution. However, owing to its excellent chemical structural stability, polyethylene exhibits well water resistance.

Revised Fig. 1. Scalable, continuous, and rapid preparation of high-performance paper plastic based on a transformative microwave radiation driven dynamic carbamate chemistry. a, Schematic demonstrating the preparation of the paper plastic. Firstly, the mixed solution of cyclic carbonate and PEI was compounded with paper, and then it was rapidly cured into paper plastic under microwave radiation. **b,** Large-volume cyclic carbonates, PEI, and mixture solution can be obtained. **c,** A large-scale sheet of paper plastic. **d,** Radar plots comparing the performance of paper plastic, paper, and polyethylene, in which the results are normalized by the maximum value of each characteristic.

References:

1. Appendix A - Data for Engineering Materials. In: *Materials Selection in Mechanical Design (Fourth Edition)* (ed Ashby MF). Butterworth-Heinemann (2011).
2. Zhou Y, *et al.* A printed, recyclable, ultra-strong, and ultra-tough graphite structural material. *Materials Today* **30**, 17-25 (2019).

3. Wang J, Emmerich L, Wu J, Vana P, Zhang K. Hydroplastic polymers as eco-friendly hydrosetting plastics. *Nature Sustainability* **4**, 877-883 (2021).
4. Li DH, *et al.* Ultrastrong, Thermally Stable, and Food-Safe Seaweed-Based Structural Material for Tableware. *Advanced Materials* **35**, 2208098 (2022).

4. According to the references below and numerous other publications, the CO₂ emission factor for PE is ~2.8 kgCO₂/kg, and for PLA, it is ~0.5 kgCO₂/kg. Supplementary Fig. 33 and related content are erroneous and must be corrected (<https://www.climatiq.io/data/emission-factor/d6dd4eb8-0b64-4d04-af59-63cec115f6d> <https://www.totalenergies-corbion.com/news/low-carbon-footprint-of-pla-confirmed-by-peer-reviewed-life-cycle-assessment/>)

Reply to the Reviewer: We thank the Reviewer #2 for the thoughtful suggestions. The Reviewer is correct that the calculated carbon footprint values in Supplementary Fig. 39 overestimate the actual footprint values. We have reviewed our calculations using the latest available OpenLCA software (2.4.0, released January 2025), the latest available ecoinvent database (ecoinvent v3.11 Cut-Off Unit Processes, released December 2024) and the Environmental Footprint 3.1 methodology. The new results are reported as Supplementary Fig. 39, and are closer to the values suggested by the Reviewer and online data (e.g. <https://www.recyclingtoday.com/news/recycled-pp-hdpe-lower-carbon-footprint-pet/>). Besides, it should be noted that the lower impact for PLA provide by the Reviewer originate from the biobased TotalEnergies Corbion PLA, while in our case, the data available corresponds to NatureWorks PLA. The data provided in Supplementary Fig. 39 consider a *cradle-to-grave* system boundary, where the manufacturing is shown in orange, and *end-of-life* are shown as a function of the footprint from incineration and fossil CO₂ emission. According to the Reviewer #2's comments, we have corrected Supplementary Fig. 39 in the revised Supplementary Information.

Supplementary Fig. 39 | Comparison of the *climate change* potential of paper plastic to a variety of benchmark plastic films and composite materials according to life cycle assessment (LCA). Left: climate change per 1 kg of material. Right: climate change normalized to tensile strength.

Reviewer #3:

Comments:

This work reports the preparation of paper plastic based on cellulose and non-isocyanate polyurethane (NIPU) by utilizing microwave radiation induced curing. The curing is claimed fast, delivering a paper plastic bearing dynamic carbamate bonds. The mechanical properties, solvent resistance, oxygen barrier properties, biocompatibility, biodegradation and recycling ability are assessed. The authors are also reporting Life-cycle assessment (LCA) and technico-economic analysis (TAE) of the process/product.

The paper is interesting, very well presented and illustrated, and easy to read. However, the degree of innovation is very limited to justify publication in Nat. Commun. The authors are introducing “paper plastic” but this is nothing else than a cellulose-based thermoset composite prepared by depositing a formulation composed of two cyclic carbonate products and a polyamine (PEI) on a cellulose paper, followed by curing. Such type of cellulose/NIPU composites have already been prepared and reported (by utilizing a more classical thermal curing), and presented also excellent mechanical properties. They were also reprocessed by exploiting the dynamic behavior of the carbamate groups (see <https://doi.org/10.1016/j.compositesa.2024.108311> for flax/NIPU composites). Moreover, although the manuscript is very well written and presented, it is oversold on many aspects by comparing the performance of the material to conventional plastics that are thermoplastics, and not thermosets nor composites. The comparison is clearly not appropriate (see detailed comments below). More appropriate comparisons should be done, with conclusions that will certainly be different to the present ones... Some info are also missing to permit the reader to reproduce the experiments.

Reply to the Reviewer: We thank the Reviewer #3 for the valuable suggestions and positive feedback, especially on our interesting, very well presented and illustrated work. We apply NIPU prepolymers to cellulose paper to achieve a coordinated and optimal development between structure and performance through dynamic bond exchange and the formation of strong hydrogen bond interactions between carbamate bonds and hydroxyl groups. While we recognize that the developed paper plastic does not hold absolute superiority in all properties, we would like to clarify that the central innovation lies in achieving an optimal balance between mechanical properties, recyclability, and water resistance—several traditionally conflicting characteristics in the field of paper-based materials as alternatives to plastic. Although there has been work on the compounding of NIPU with cellulose materials, there is still a clear difference between this type of work and the preparation of paper plastics. To illustrate the novelty of our study, the representative work (<https://doi.org/10.1016/j.compositesa.2024.108311> for flax/NIPU composites) are selected for detailed discussion. Our work differs from previously reported work as described below.

<https://doi.org/10.1016/j.compositesa.2024.108311> for flax/NIPU composites

Abstract

We herein propose capitalizing on strong hydrogen bonding from novel bio-CO₂-derived dynamic thermoset to achieve high-performance natural fiber composites (NFC) with circular features. CO₂- and biomass-derived polyhydroxyurethane (PHU) thermosets were selected, for the first time of our knowledge, as matrices for their ability to make strong H-bond, resulting in outstanding mechanical properties for NFC. Exploiting this H-bond key feature, exceptional interface bonding between flax and PHU was confirmed by atomic force microscopy and rationalized by atomistic simulation. Without any treatment, an increase of 30% of stiffness and strength was unveiled compared to an epoxy benchmark, reaching 35 GPa and 440 MPa respectively. Related to the thermoreversible nature of hydroxyurethane moieties, cured flax-PHU were successfully self-welded and displayed promising properties, together with recyclability features. This opens advanced opportunities that cannot be reached with epoxy-based composites. Implementing CO₂-derived thermosets in NFC could lead to more circular materials, critical for achieving sustainability goals.

Fig. 1. (a) Formation of cyclic carbonate by addition of CO₂ to epoxy, (b) Epoxy-amine representative network, and (c) PHU representative network. (Full line insets represent the network crosslinking node, and dotted line insets represent the potential inter-/intra-molecular H-bond formed.).

In this work (<https://doi.org/10.1016/j.compositesa.2024.108311>), a novel approach was introduced to developing high-performance, circular natural fiber composites (NFCs) using CO₂- and biomass-derived polyhydroxyurethane (PHU) thermosets as sustainable matrices. By leveraging the strong hydrogen-bonding capability, the researchers achieved exceptional interfacial adhesion between flax

fibers and the PHU matrix, resulting in a flax-PHU composites that exhibits remarkable mechanical properties. Additionally, the thermoreversible nature of hydroxyurethane bonds enabled self-welding and recyclability of the cured composites—features unattainable with traditional epoxy systems. This work highlights the potential of CO₂-derived PHU thermosets to advance sustainable materials by combining high performance with circularity, addressing critical environmental goals.

Our work (high mechanical properties, biodegradable and recyclable Paper Plastic)

Abstract

In response to the looming concerns of plastic pollution, replacing plastic with paper is a very promising way, but its realization seems a long way off due to the poor water resistance and unsatisfied mechanical strength of cellulose fibril-based materials. Herein, we develop a versatile functionalizing material consisting of mainly biobased cyclic carbonate-bearing compounds and amine compound, which can enable the rapid transformation (within 2 min under microwave radiation) of the cellulose paper into plastic-like material (named paper plastic) having an unprecedentedly high tensile strength of ~126 MPa. Through a systematic experimental and theoretical study, the paper plastic's unique combination of excellent mechanical properties and water/solvent resistance is attributed to the easy formation of carbamate abundant non-isocyanate polyurethane cooperated with the intermolecular bond exchange mechanism between the dynamic carbamate moiety and hydroxyl of the cellulose. Also, benefiting from the high content (>80%) and natural advantages of biobased materials, the paper plastic shows superior thermal stability, processability, and biodegradability than most petrochemical-based plastics, promising the great potential of dynamic carbamate chemistry toward high-performing paper plastic composites.

Fig. 2. Design of the paper plastic by mixing cyclic carbonate with amines and applying them to the paper.

In our work, we propose a simple and easy-to-operate method for the rapid preparation of high-strength, water- and solvent-resistant, biocompatible, biodegradable, and recyclable paper plastics based on microwave radiation-driven carbamate dynamic chemistry. Microwave radiation can promote the curing reaction to be completed within 2 minutes, which greatly shortens the reaction time thus reducing energy consumption and carbon footprint; meanwhile the prepared paper plastic still shows high tensile strength (126 MPa) of isotropy. Paper plastics exhibit a high bio-based content of 82%, which is a great advantage for green development. Meanwhile, the high biological content is also conducive to the realization of biocompatibility and biodegradability, outperforming most petroleum-based plastics. The excellent water/solvent resistance and mechanical robustness of paper plastics are due to the synergistic effect of carbamate-rich non-isocyanate polyurethanes forming strong hydrogen bonds with paper and the dynamic bond exchange between the urethane moiety and the cellulose hydroxyl group. The dynamic carbamate chemistry not only enhances performance of paper-based materials but also opens pathways for eco-friendly, high-strength paper plastic composites.

To illustrate the originality of our current work, we summarize the major differences between our work and the reference provided by Reviewer #3, as shown below.

1. **Microwave radiation curing (Our work) VS Traditional thermal curing (Reference).** The simplification of the curing process is an important process to simplify the steps of material preparation and improve the production capacity. Microwave radiation can promote the curing reaction to be completed within 2 minutes, which greatly shortens the reaction time, but the prepared paper plastic still shows tensile strength, water resistance, and other performances. The flax fiber composites prepared by traditional thermal curing methods in the references also have excellent properties. However, the curing process requires multiple steps, preheating at 60 °C, 2 hours at 80 °C, 1 hour at 100 °C, and finally 1 hour at 150 °C for post-curing.
2. **Applicable to any plain paper substrate with excellent processability (Our work) VS Applicable to orientally arranged flax fibers (Reference).** To demonstrate the universality of the paper plastic preparation strategy we proposed, we selected various untreated types of paper (including printing paper, filter paper, and kraft paper, among others), applied a NIPU prepolymer onto their surfaces, and subsequently cured them to obtain the desired paper plastic composites. Paper plastics with different paper bases have shown significant improvements in mechanical properties, indicating that our preparation method is simple and does not require restriction of raw materials. Moreover, our paper plastic could be made into different types of products including straws, bags, and even thick board, demonstrating excellent processability. However, only flax fibers with high specific mechanical properties were chosen as substrates in the references. It remains unclear whether other types of substrates are also applicable to their approach. And the processability of flax fiber reinforced composites into various types of products also remain unexplored.

3. **Higher cellulose paper usage, better environmental friendliness..** Based on our previous work, we chose NIPU mixtures (cyclic carbonate compounds and amine compounds) with a mass ratio of 1:1 to paper for the preparation of paper plastics. But in the reference, the epoxy-amine or CC-amine mixture were prepared by weighting about 1.4 times the mass of fibers. The use of fewer compounds not only reduces costs, but also facilitates the rapid biodegradation of materials due to the high bio-based cellulose fiber content.
4. **Isotropic mechanical strength (Our work) VS Anisotropy mechanical strength (Reference).** In our work, the strength of the base NIPU polymer is only a few MPa, but after modification, a tensile strength greater than 100 MPa can be obtained, which exhibits a significant strength improvement of tens of times. Meanwhile, since the fiber structure of the ordinary paper we use is not oriented to each other, the resulting paper plastic is isotropic, and we have also verified the isotropy of the material through SAXS testing. However, the strength of the base polymer PHU in the reference can reach about 80 MPa, which is significantly better than the base polymer we use, so it should be understandable that the strength of the composite polymer can reach more than 400 MPa. In addition, since the initial flax fibers are oriented, the resulting composite has a longitudinal tensile strength of 440 MPa and a transverse tensile strength of only 39 MPa, which may limit the application in the field of transverse stress materials.
5. **Exploration of the mechanism of structure formation and mechanical property enhancement (Our work).** We analyze the mechanism by which the structure affects the mechanical enhancement of materials through simulation calculations. By simulating the chemical cross-linking and physical interaction between the molecular network and the cellulose paper, the mechanical properties obtained by the simulated tensile test are close to the results of the actual test. This result also conversely confirms that the carbamate bonds in NIPU react with the cellulose in the paper to form interactions (including chemical cross-linking and physical winding interactions). We also used solid-state NMR to analyze the structure of cellulose paper and paper plastics to indicate the formation of chemical and hydrogen bonds between carbamate bonds and cellulose. Through these analyses, we can better illustrate the structure of cellulose paper after modification and the mechanism of mechanical strength enhancement. The gained insights into the structure formation and structure-mechanical property relationship could guide composite materials and plastic alternatives design. The advanced characterization and simulation results collectively reveal the strengthening mechanism of our paper plastic that is substantially different from the Flax-PHU composites reported in the reference paper.
6. **Packaging products demonstration and comprehensive performance evaluation (Our work).** Our work explores the comprehensive properties of paper plastics and packaging products, including mechanical properties, thermal stability, dimensional stability, water and solvent resistance, biocompatibility, biodegradability, gas barrier, recycling and reprocessing capabilities, as well as life cycle assessment (LCA), among others. Meanwhile, we have also carried out large-scale material preparation to illustrate the simplicity and ease of operation of the paper plastic

preparation process. This comprehensive analysis is critical for plastic alternative used in packaging field, which, however, are mostly absent in the reference paper.

In brief, we report a microwave radiation driven chemical modification strategy to rapidly transform (within 2 min) cellulose paper into plastic-like paper material with substantially improved mechanical strength and water resistance based on dynamic carbamate chemistry. Paper plastics were prepared by using bio-based cyclic carbonate compounds with CO₂ immobilization ability and amine compounds to chemically modify the cellulose in the paper network, in which the dynamic carbamate bond of NIPU structure can promote the rearrangement of the network structure through bond exchange reaction with the hydroxyl group (–OH) in the cellulose paper, and can also form strong hydrogen bonds to improve the mechanical strength. The result is a solution to several traditional problems with paper-based materials as an alternative to plastics, achieving the best balance between mechanical properties, recyclability, biodegradability, and water resistance.

Moreover, to give a clearer picture of our innovative work, we summarize the novelty of our work as follows:

(a) Microwave radiation curing is introduced to optimize the preparation process

A novel microwave radiation driven dynamic carbamate chemistry approach to rapidly transform (within 2 min) regular paper into high-performing paper plastic with substantially improved overall performances, including mechanical properties, water and solvent resistance, and thermal stability. Rapid formation of cross-linked dynamic network between the NIPU and cellulose fibers in paper occurs upon microwave radiation. As a result, the dynamic carbamate bond in NIPU structure facilitates the rearrangement of network structure by bond exchange reaction with hydroxyl group of the cellulose paper, contributing to substantial improvements of overall performances of the paper plastics compared with paper.

(b) Clarified the combination mode and mechanical property enhancement mechanism between polymer and cellulose paper

Through advanced analytical techniques (NMR spectra, Raman spectra, etc) and theoretical simulation, we confirmed that NIPU not only fills the pore structure of cellulose-based paper to form physical reinforcement, but also forms efficient chemical cross-linked dynamic network and hydrogen bond between NIPU and cellulose. These combined factors have resulted in remarkable enhancements in the strength and other properties of paper-based materials. Paper plastics constructed from different kinds of paper (such as printing paper, toilet paper, filter, etc.) and NIPU all show significant improvement in mechanical properties, highlighting the universality of our approach. Furthermore, the preparatory efficacy of paper-based alternatives such as paper straws, paper cups, paper bags and cardboard made from paper plastics was also assessed. We anticipate that these findings will foster the accelerated and enhanced development and implementation of the plastic replacement industry.

(c) The design concept embraces environmental sustainability, aligning with the current demands of global carbon reduction

The main raw materials employed in the fabrication of paper plastics encompass biomass resources, such as cellulose, oil, and turpentine. The experimental test conducted by Beta Analytic Inc. reveals a bio-based carbon content of 82%, surpassing the requirements of numerous international bio-based labels. Moreover, the synthesis of cyclic carbonate for paper plastics preparation utilizing CO₂ as a raw material holds significant implications for achieving carbon neutrality objectives. The paper plastic also demonstrates excellent biodegradability, biocompatibility, and recyclability. The analysis of LCA results also showed that the preparation process of paper plastics produced a low carbon footprint. With these combined advantageous features, we believe our proposed strategy in transforming regular paper into paper plastic can be regarded as one of the guidelines for the future development of sustainable materials.

In addition, we appreciate Reviewer #3's constructive suggestions and comments regarding the current manuscript. At the same time, we are aware that the academic quality of our manuscript should be further improved to dispel these doubts, and that is what we've been working on in the past few months. While addressing all the comments, we have further improved this work given more comprehensive experiment (Figure R1). **The experimental and computational efforts main in the Revised Manuscript are summarized below:**

I. Supplement the information on raw materials and the detailed description of each step in the preparation of paper plastics

We describe the raw materials and the conditions of the preparation process of paper plastics in more detail, in order to provide recommendations for the subsequent preparation of such paper plastics. The supplementary preparation process information includes the structure characterization, molecular weight, raw material ratio, viscosity change of NIPU prepolymers, and the method of compounding NIPU prepolymer with cellulose paper. This information completes the method of mass production of paper plastic and explains why the side reaction of urea formation occurs almost non-existent in the preparation of paper plastic.

II. Inclusion of a comparative study of performance between the paper plastic and other conventional thermoset materials

To complete our performance analysis, the thermal dimensional stability, coefficient of thermal expansion (CTE), and oxygen resistance of thermoset materials were compared. The paper plastic is supplemented by an extended water contact angle over time and a change in size after immersion in a solvent, further analysis of water and solvent resistance.

III. Optimization of the Life Cycle Assessment (LCA)

In order to make a more accurate analysis of the life cycle assessment of paper plastics, we invited Dr. Erlantz Lizundia, an expert in the field of LCA. The manufacturing of paper plastic (cradle-to-gate)

presents a carbon footprint of 6.81 kg CO₂ equiv kg⁻¹. Due to the simplicity of the manufacturing process, significant bio-based carbon content, and the high tensile strength of paper plastics, the climate change potential (normalized to tensile strength) shows a significantly lower value for paper plastic over 14 potentially competitive benchmark plastic films and composite materials.

IV. Optimization of biodegradability testing

We supplemented the quantitative analysis of the biodegradation process of paper plastics using biological compost testing. The amount of CO₂ released during the degradation process of paper plastics was recorded, and based on the CO₂ emissions data, it was calculated that paper plastics could achieve a degradation rate of 45% after 45 days of composting. We also utilized the pyrolysis-gas chromatography-mass spectrometry (Py-GCMS) technique to investigate the possibility of transforming into microplastics during the degradation process. The results show that paper plastics do not produce microplastics during degradation.

V. Further demonstration of the recyclability (achieved by dynamic carbamate chemistry) of the NIPU in paper plastic

To better illustrate the dynamic performance of the network structure and the mechanism of recycling capability, we supplemented the reprocessing experiments of NIPU. Through the exploration of physical hot pressing reprocessing, chemical degradation recycling, scratch self-healing, and the mechanical properties before and after reprocessing, it is shown that NIPU in paper plastic has good network reorganization capability. Therefore, this also contributes to the realization of the recycling performance of paper plastic.

I. Inclusion of the information on raw materials

II. Inclusion of a comparative study of performance between the paper plastic and other conventional thermoset materials

III. Optimization of the Life Cycle Assessment (LCA)

IV. Optimization of biodegradability testing

V. Recyclability of the NIPU in the paper plastic

Fig. R1 Summary of our experimental efforts during the past few months to provide more solid evidences towards the excellent mechanical properties and environmental friendliness of the developed paper plastics, and provides a deeper understanding of the structure-performance-function relationship.

Selected comments:

1. - *Fig 2a and related discussion. What is the NH₂ functionality of PEI? What is the molar mass of PEI? There are many different PEI and the authors should give these information to allow the reader to reproduce the experiments.*

Reply to the Reviewer: We thank the Reviewer #3 for the constructive comments. The NH₂ functional group in PEI reacts with the cyclic carbonate group to form a carbamate bond, resulting in a polymer non-isocyanate polyurethane. The molecular weight of PEI used in our work is 1800 g/mol, which is illustrated in the manuscript "Materials and chemicals".

2. - *Fig 2d and e: the color code is not defined*

Reply to the Reviewer: We thank the Reviewer #3 for the thoughtful comments. We have modified Figs. 2d and e to clarify what the color code represents in the revised manuscript. The green curve represents paper plastic, while the blue curve represents pure paper.

Revised Fig. 2 | Morphological and structural features of paper plastic. a, Schematic of polymer (NIPU) preparation. **b**, FTIR spectra of CSBO, APAC, PEI, and NIPU. **c**, SEM images of cellulose paper and paper plastic. **d**, One-dimensional Raman spectra and two-dimensional (2D) Raman spectra constructed from the intensity of C=O. **e**, FTIR spectra of paper and paper plastic.

3. - Page 7. “A molar ratio of CSBO: (APAC + PEI) = 1: 1 is adopted to theoretically ensure the complete reaction of cyclic carbonate 125 and amino moieties to form carbamate dynamic bonds... ». The molar ratio used is absolutely unclear. If this ratio is used, there will be an excess of cyclic carbonate coming from CSBO – but it will depend on the functionality of PEI that is not mentioned in the document. Do the authors discuss about molar ratios of the molecules or the functional groups?
 – this is not the same. For clarity to the reader, they should express the molar ratios of the functional groups, thus of the cyclic carbonate and primary amine groups.

Reply to the Reviewer: We thank the Reviewer #3 for the constructive comments. We agree with the reviewer's proposal to use the molar ratio of functional groups to illustrate the proportion of raw materials added. In fact, we do use the molar ratio of functional groups (cyclic carbonate group: amino group = 1:1) to calculate the amount of raw materials added in the preparation of materials. The "Fabrication of Paper Plastics" section in the manuscript details the quantities of each raw material

utilized. To facilitate a more comprehensive discussion of these quantities, Supplementary Tab. 1 has been appended to present this information clearly. Additionally, the description in the revised manuscript has been updated accordingly.

Supplementary Table 1 | Formulations of the raw materials.

Raw materials	Molar mass (M _w , g/mol)	Content (g)	Cyclic carbonate groups content (mol)	Amino groups content (mol)
APAC	574	50	0.174	-
CSBO	1122	310	1.105	-
PEI	1800	140	-	1.29
The molar ratio of cyclic carbonate group and amino group is about 1:1				

Updates to the revised manuscript: According to Reviewer #3’s comments, we have changed the description of the raw material molar ratio in the revised manuscript on Page 7.

“A molar ratio of 1:1 between the cyclic carbonate groups (from CSBO and APAC) and the amino groups (from PEI) is employed to theoretically ensure the complete reaction of these functional groups to form carbamate dynamic bonds that are highly attractive for mechanical strength and bond exchangeability.”

4. - Page 9. *“we reasoned that the reformation of carbamate moiety between cellulose and NIPU is responsible for the significant interfacial reinforcement and therefore the mechanical properties. » There is no convincing data in the manuscript proving the formation of covalent bonds between the cellulose and the NIPU by transurethanization. Hydrogen bonding between NIPU and the cellulose might also account for the interfacial and mechanical properties reinforcement as it has been shown in <https://doi.org/10.1016/j.compositesa.2024.108311>. This should be discussed in regards to this seminal work.*

Reply to the Reviewer: We thank the Reviewer #3 for the valuable suggestions. We agree that the hydrogen bonding proposed by the reviewer also has a positive effect on the mechanical strength enhancement of paper plastics. We also mentioned in the manuscript that the absorption peaks value of the functional groups in the FTIR spectrum indicates the existence of hydrogen bonds between carbamate bonds (*“Furthermore, the presence of hydrogen bonds between carbamate groups or between carbamate groups and hydroxyl groups leads to the emergence of new absorption peaks in the spectrum of paper plastic within the range of 1715 to 1695 cm⁻¹ when compared with paper.”*), but we may not describe it enough in the manuscript. Therefore, in line with the reviewers' recommendations, we have added a clearer expression that hydrogen bonding between NIPU and cellulose may also be the cause of the enhancement of interfacial and mechanical properties.

In addition, we investigated the reaction of carbamate bonds with hydroxyl groups on cellulose chains by comparing ^{13}C NMR spectra of paper plastics with pure paper. The new peak at 153.7 ppm attributed to the carbamate group, peak shift of C1 (107.3 ppm) and C4 (90.8 ppm) of cellulose to the high field and a significant shoulder peak of C6 (65.4 ppm) were observed, which is due to the cross-linking of the C6–OH with NIPU network (Supplementary Fig. 10). The interaction between cellulose and NIPU was also modelled, and theoretical calculations showed that the formation of carbamate linkages between NIPU and cellulose had the thermodynamic advantage of high binding energy (Supplementary Table 2). In the case of simulated carbamate bond cross-linking and hydrogen bonding between cellulose and NIPU, the tensile strength obtained by the simulation is close to the experimental test results, which also indirectly confirms the formation of chemical bonds between cellulose paper and NIPU. Therefore, in order to better describe the reasons for the improvement of mechanical properties, we have modified the corresponding description in the manuscript.

Updates to the revised manuscript: To enrich the analysis of the performance enhancement mechanism between NIPU and cellulose paper, we have added more careful descriptions and reference in the revised manuscript on Page 11.

“Considering the low tensile strength of only ~3.2 MPa for NIPU, it is reasoned that the reformation of the hydrogen bonds and carbamate linkages between cellulose and NIPU is responsible for the significant interfacial reinforcement and therefore the mechanical properties.”

5. - Page 12. Comparison of the CTE values and softening/shrinkage of paper plastic when placed at 150 °C with those of commercial plastics (PP, PE, PS, PLA). This comparison is not appropriate as the authors compare thermoplastics (PP, PE, PS, PLA) to a thermoset composite (the paper plastic). This is well-known that thermosets have improved thermal stability compared to thermoplastics! A fair comparison would be to compare the paper plastic with a similar paper plastic made from another thermoset, for instance an epoxy resin or even, a conventional PU thermoset. The message will be totally different...

Reply to the Reviewer: We appreciate the Reviewer #3 for the critical insight regarding the thermal performance comparison between our paper plastic and conventional thermoplastics. While the initial comparison with PP/PE/PS/PLA aimed to contextualize our material’s potential as a sustainable alternative to plastic packaging (where thermoplastics dominate the market), we fully agree that contrasting paper plastic with thermoplastics in thermal stability may oversimplify the discussion. Therefore, we provide a revised analysis and additional thermoset composite data to address this concern. We purchased commercial epoxy resins and polyurethanes (PU) from Jiangsu Zhongtian Rubber & Plastic Company and performed thermal expansion tests using a thermomechanical analyzer (TMA Q400). The coefficient of thermal expansion (CTE) of the materials was also recorded from -20 °C to 180 °C. In Supplementary Fig. 23, the epoxy exhibits good dimensional stability and

a low coefficient of thermal expansion (CTE, $36 \mu\text{m}/(\text{m}\cdot^\circ\text{C})$) as the temperature changes. Paper plastic has similar thermal dimensional stability with epoxy resin due to the stability of the paper structure. PU exhibits less ideal thermal stability, which is manifested in significant dimensional changes during temperature changes and fractures in the middle of the process. After heating epoxy resin and PU in an oven at 150°C for 20 min (Supplementary Fig. 23c), the size of epoxy did not change visibly due to its excellent thermal stability, while PU showed curling deformation. These results show that paper plastics can exhibit thermal stability similar to thermosetting epoxy resins.

Supplementary Fig. 23 | Thermal stability of epoxy and polyurethanes (PU). a, Coefficient of thermal expansion (CTE). b, Thermal expansion. c, Dimensional stability experiment at 27°C (R.T.) and 150°C .

Updates to the revised manuscript: According to Reviewer #3’s comments, we have added additional descriptions of the thermal stability of epoxy and PU in the revised manuscript on Page 15 and Supplementary Information.

“For better comparison with commercial plastics, we have also supplemented the thermal expansion test results of epoxy and polyurethane (PU) (Supplementary Fig. 23). These results show that paper plastics can exhibit thermal stability similar to thermosetting epoxy resins, and further prove that paper plastics have well thermal stability.” on Page 15 in the revised manuscript.

“We purchased commercial epoxy resins and polyurethanes (PU) from Jiangsu Zhongtian Rubber & Plastic Company and performed thermal expansion tests using a thermomechanical analyzer (TMA Q400). The coefficient of thermal expansion (CTE) of the materials was also recorded from -20°C to 180°C . In Supplementary Fig. 23, the epoxy exhibits good dimensional stability and a low coefficient of thermal expansion (CTE, $36 \mu\text{m}/(\text{m}\cdot^\circ\text{C})$) as the temperature changes. Paper plastic has similar thermal dimensional stability with epoxy resin due to the stability of the paper structure. PU exhibits less ideal thermal stability, which is manifested in significant dimensional changes during temperature changes and fractures in the middle of the process. After heating epoxy resin and PU in an oven at 150°C for 20 min (Supplementary Fig. 23c), the size of epoxy did not change visibly due to its excellent thermal stability, while PU showed curling deformation.” in the Supplementary Information.

6. - Page 13. *“The surface wettability test revealed that the paper plastic demonstrates exceptional hydrophobicity, as evidenced by a contact angle of over 100° (Fig. 4c), in contrast to that water droplets on the surface of the paper are rapidly absorbed, showing a water of ~160%. » Such type of phrasing oversells the technology. It is evident that coating the hydrophilic cellulosic-based paper by a hydrophobic NIPU will strongly increase the contact angle to the value of NIPU (100.13 °, suppl Fig 23). Nothing exceptional here. Moreover, did the authors follow the evolution of the contact angle with time? NIPUs absorb water due to the pending OH groups and the contact angle generally decreases over time.*

Reply to the Reviewer: We sincerely appreciate the Reviewer #3 for insightful comments regarding the characterization of surface wettability. We agree that careful interpretation of contact angle data is critical, and we have revised the manuscript accordingly to better express the preparation effect of our paper plastics. We also investigated the contact angle with time of paper plastics to illustrate their hydrophobicity.

Regarding the comment “The surface wettability test revealed that the paper plastic demonstrates exceptional hydrophobicity, as evidenced by a contact angle of over 100° (Fig. 4c), in contrast to that water droplets on the surface of the paper are rapidly absorbed, showing a water of ~160%. » Such type of phrasing oversells the technology. It is evident that coating the hydrophilic cellulosic-based paper by a hydrophobic NIPU will strongly increase the contact angle to the value of NIPU (100.13°, suppl. Fig 23). Nothing exceptional here.”:

According to the suggestion of Reviewer #3, we modified the description of "excellent hydrophobicity" in the revised manuscript to more accurately express the hydrophobicity of paper plastics. Actually, within the scope of our work, we aim to highlight that the innovation of this research does not lie in the enhancement of hydrophobicity alone, but rather in achieving such performance while preserving a high bio-based content and recyclability. Therefore, we have adjusted the wording in the revised manuscript to avoid the word "exceptional" and instead emphasize the comprehensive properties of the materials.

Updates to the revised manuscript: According to Reviewer #3’s comments, we have added a more careful description in the revised manuscript on Page 17.

“The NIPU-coated paper exhibits a water contact angle of over 100° (Fig. 4c), demonstrating significant hydrophobicity compared to unmodified paper (almost no contact angle). Taking into account the influence of hydroxyl groups in the paper plastic, we supplemented the evolution of the contact angle over time. As shown in Supplementary Fig. 26, the contact angle of the paper-plastic surface hardly changes as the time extends from 0 to 120 min, and remains at about 100°, which indicates that the stable hydrophobicity and suggests that the crosslinked NIPU network effectively restricts water penetration. Moreover, Paper and paper plastics are immersed in

water for 7 days and their water absorption is recorded. Figure 4c shows that paper exhibits 160% water absorption, while paper plastic absorbs only 20%. The wet tensile strength of the paper plastic can be maintained at 85 MPa (about 72% of the dry paper plastic) after soaking for one week. These results indicated that the incorporation of NIPU into the paper network effectively enhances the mechanical properties of paper plastic in wet environments. Importantly, this hydrophobicity is achieved while maintaining high bio-based content (82%) and full recyclability, which represents a key advance over conventional petroleum-based plastic coatings.”

Regarding the comment “Moreover, did the authors follow the evolution of the contact angle with time? NIPUs absorb water due to the pending OH groups and the contact angle generally decreases over time.”:

Taking into account the influence of hydroxyl groups in the paper plastic, we supplemented the evolution of the contact angle over time based on the comments of the reviewer. As shown in Supplementary Fig. 26, the contact angle of the paper-plastic surface hardly changes as the time extends from 0 min to 120 min, and remains at about 106°, which indicates that the stable hydrophobicity. This result suggests that the crosslinked NIPU network effectively restricts water penetration.

Supplementary Fig. 26 | Analysis of water and solvent resistance. a, Water contact angle of NIPU. b, Change in water contact angle of paper plastic over time. c, Images of the paper plastic after being immersed in different organic solvents for 7 days.

Updates to the revised manuscript: According to Reviewer #3’s comments, we have added a more careful description about the changes of the contact angle over time in the revised manuscript on Page 17.

“Taking into account the influence of hydroxyl groups in the paper plastic, we supplemented the evolution of the contact angle over time. As shown in Supplementary Fig. 26b, the contact angle of the paper-plastic surface hardly changes as the time extends from 0 min to 120 min, and remains at about 106°, which indicates that the stable hydrophobicity and suggests that the crosslinked NIPU network effectively restricts water penetration.”

7. - Pages 13-14. *“The state of paper plastics was found to be less than 20%, leading to a high wet-state tensile strength of 85 MPa (about 72% of dry paper plastic), still surpassing some commercial plastics' tensile strength in dry environments ». Comparison is here not appropriate too. Prepare a similar paper plastic with another thermoset instead of thermoplastics, the message will be different.*

Reply to the Reviewer: We thank Reviewer #3 for the constructive comments, especially for point regarding the comparison of mechanical performance. What we would like to illustrate is that the motivation for the initial comparison of our work with thermoplastics (e.g., PE, PP) is that our material targets are applied to the packaging sector, which has traditionally been dominated by these polymers. Our paper plastics exhibit similar or slightly better mechanical strength to these materials. Meanwhile, our comparison here is to emphasize that paper plastic has good water resistance, and still has a tensile strength of 85 MPa after being immersed in water for a week. For some applications of plastic packaging products that require a tensile strength of about 85 MPa, our paper plastics will not be limited in application, and still have the potential to replace traditional thermoplastics used in this field.

Nonetheless, we fully agree with Reviewer #3 that it is important to not only compare with thermoplastics, but also with thermosets. To address this concern, we compared our paper plastic with commercial thermosets. Thermoset plastic epoxy resins offer high hardness, strength, and rigidity, with typical tensile strengths ranging from 40–100 MPa. The strength of polyurethane is 25–51 MPa, and the strength of polyester is 41–90 MPa. While thermosets may achieve absolute dry strength, their typical lack of recyclability (e.g., permanent crosslinking) conflicts with our goal of closed-loop recycling. Our material starts with cellulose-based paper substrate modified through green in-situ polymerization of bio-derived monomers. Traditional paper lacks solvent resistance, exhibits poor wet strength, even disintegrates in polar solvent, and is extremely limited when used as a packaging material. Our design intentionally preserves the biodegradable backbone of cellulose while introducing solvent resistance through a dynamically crosslinked network that balances mechanical properties, solvent stability, and degradability.

Updates to the revised manuscript: Thank you for emphasizing the mechanical performance comparisons. We have revised the manuscript to clarify the hydrophobicity and wet strength.

“The wet tensile strength of the paper plastic can be maintained at 85 MPa (about 72% of the dry paper plastic) after soaking for one week. These results indicated that the incorporation of

NIPU into the paper network effectively enhances the mechanical properties of paper plastic in wet environments.”

8. - Page 14. *“These results indicated that the incorporation of NIPU into the paper network effectively enhances the mechanical properties of paper plastic in wet environments. » This is evident that an hydrophobic coating of NIPU on the hydrophilic paper will enhance its stability in water, and thus its mechanical properties. Nothing new and surprising.*

Reply to the Reviewer: We sincerely appreciate the reviewer's perspective on the role of hydrophobic coatings in enhancing paper stability. While surface hydrophobization is indeed a known strategy, we respectfully clarify that our approach fundamentally differs from conventional coating techniques in both mechanism and outcome. The novelty lies in the multi-scale structural integration of NIPU within the cellulose paper matrix, rather than mere surface modification.

Structural design that is different from existing coating technologies. Traditional hydrophobic coatings (e.g., wax, PE lamination) form an impermeable barrier on paper surfaces through physical deposition. Our methodology enables in-situ polymerization of NIPU monomers within the cellulose fiber. Simultaneously, the carbamate bonds generated by these monomers react with the hydroxyl groups present in the cellulose, thereby forming a cross-linked network structure (including hydrogen bond interactions and chemical bond cross-linking). This multiple integration mechanism leads to enhanced water resistance and improved mechanical properties of cellulose paper.

Achieving a sustainable and multi-functional equilibrium. Traditional hydrophobic coatings create unrecyclable waste, whereas our designs are highly bio-based (82%), closed-loop recyclable, and biodegradable. Paper plastics achieve hydrophobicity while maintaining environmentally friendly properties that cannot be achieved with conventional coatings. In addition, NIPU monomer can enhance the stability of the overall structure by forming a network of cross-links with cellulose. This not only improves the water resistance of paper-based materials and enhances their mechanical strength but also enables the preparation of various products (e.g., bags, cardboard) through the dynamic characteristics of carbamate bonds.

9. - Page 14. *Hot pressing of the paper plastic to form the bag. Exploiting the dynamic behavior of the carbamate bonds to reprocess NIPU thermosets as well as NIPU composites is not new. If the reviewer reads well, there is no experimental section dealing with the procedure used for producing this bag. What is the pressure applied, the contact time and, importantly, the temperature? Generally, NIPUs require a quite high temperature to be reprocessed (160 °C) that induces side reactions such as urea formation that limits further reprocessing steps. Did the authors observe some urea formation (IR is a simple tool to monitor them) and if no, why?*

Reply to the Reviewer: We sincerely appreciate the Reviewer #3 for the insightful comment.

Regarding the comment “Hot pressing of the paper plastic to form the bag. Exploiting the dynamic behavior of the carbamate bonds to reprocess NIPU thermosets as well as NIPU composites is not new. If the reviewer reads well, there is no experimental section dealing with the procedure used for producing this bag. What is the pressure applied, the contact time and, importantly, the temperature?”:

We agree that dynamic covalent chemistry in NIPUs is well-established by the exchange reaction between carbamate bonds and hydroxyl groups. However, our work uniquely integrates this concept with cellulose-based paper rich in hydroxyl groups to achieve enhanced mechanical properties, closed-loop recyclability, degradability, water and solvent resistance and more. Thanks to Reviewer #3 for the attention to detail in the experiment. The edges of the paper plastic are stacked together and then hot-pressed at 120 °C and 5 MPa for 1 hours. In this hot pressing process, the carbamate bond exchange reaction was used to achieve heat sealing without adhesive between the paper-plastic films, and then the packaging bag was obtained (Fig. 5b). The description is supplemented in the preparation section of paper plastics.

Updates to the revised manuscript: According to Reviewer #3’s comments, we have added a more careful description of the preparation process of paper plastic products in the revised manuscript on Page 31.

“Paper plastic board and paper bag products are prepared by stacking paper plastics on top of each other and then pressing them at 120 °C and 5 MPa for 1 hours.”

Regarding the comment “Generally, NIPUs require a quite high temperature to be reprocessed (160 °C) that induces side reactions such as urea formation that limits further reprocessing steps. Did the authors observe some urea formation (IR is a simple tool to monitor them) and if no, why?”:

We thank the Reviewer #3 for raising this important point. Our hot pressing process uses a temperature of 120 °C, which is below 160 °C, which kinetically hinders the side reaction of urea formation. In addition, even if a trace amount of cyclic carbonate is decomposed into isocyanate, a large number of hydroxyl groups on the cellulose will react preferentially with it. The molar ratio of the amino to cyclic carbonate groups used in our work is 1:1, which also theoretically ensures that there is no free –NH₂ residues left over to form urea side reactions. Therefore, the side reaction of urea formation has little effect on the reprocessing process of paper plastics. In addition, there is a large amount of work¹⁻⁴ to explore the reprocessing performance of NIPU, which has confirmed that the presence of excess hydroxyl groups effectively promotes the ability of the material to self-healing and recycle due to the reaction of the cyclic carbonate group with amino groups to form carbamate bonds with β–OH.

We also monitored urea formation via FTIR spectra analysis according the suggestion of Reviewer #3. The C=O absorption peak of the urea and carbamate bond is similar, and since the absorption peak of cellulose also occurs around 1650 cm⁻¹, it is difficult to determine whether a urea group is generated

by the absorption peak there. The difference is that the two N–H vibrations in the urea group (–NH–C(=O)–NH–) exhibit a strong absorption peak around 3300 cm⁻¹. However, in the FTIR spectra of paper plastics (Fig. 2e), there is only a wide –OH absorption peak similar to that of pure paper, but no strong absorption band of –NH, which also indicates that the by-product urea bond formation is less or even below the detection limit or is not generated. Therefore, we should assume that the process of preparing paper bags from paper plastics will not affect its subsequent reprocessing process.

Revised Fig. 2 | Morphological and structural features of paper plastic. a, Schematic of polymer (NIPU) preparation. **b**, FTIR spectra of CSBO, APAC, PEI, and NIPU. **c**, SEM images of cellulose paper and paper plastic. **d**, One-dimensional Raman spectra and two-dimensional (2D) Raman spectra constructed from the intensity of C=O. **e**, FTIR spectra of paper and paper plastic.

References:

1. Fortman DJ, Brutman JP, Cramer CJ, Hillmyer MA, Dichtel WR. Mechanically activated, catalyst-free polyhydroxyurethane vitrimers. *J Am Chem Soc* 137, 14019-14022 (2015).

2. Gomez-Lopez A, Elizalde F, Calvo I, Sardon H. Trends in non-isocyanate polyurethane (NIPU) development. *Chem Commun (Camb)* 57, 12254-12265 (2021).
3. Maisonneuve L, Lamarzelle O, Rix E, Grau E, Cramail H. Isocyanate-Free Routes to Polyurethanes and Poly(hydroxy Urethane)s. *Chem Rev* 115, 12407-12439 (2015).
4. Lucherelli MA, Duval A, Avérous L. Biobased vitrimers: Towards sustainable and adaptable performing polymer materials. *Progress in Polymer Science* 127, 101515 (2022).

10. - Fig 4e. As the paper plastic is a composite thermoset, it is normal that the product does not dissolve in organic solvents and presents solvent resistance. This is one of the numerous advantages of thermosets.

Reply to the Reviewer: We sincerely appreciate the Reviewer's insightful comment regarding the inherent solvent resistance of thermosets. We fully agree that conventional thermosets exhibit solvent resistance due to their crosslinked network structure, thus we acknowledge the importance of clarifying the novelty of our work in this context. Please allow us to elaborate on the key distinctions between our degradable paper plastic and traditional thermosets:

Material Origin & Sustainability Focus. Traditional thermosets (e.g., epoxy, phenolic resins) are typically derived from non-renewable petrochemicals and face challenges in degradation/recycling, causing environmental persistence. In contrast, our material starts with cellulose-based paper substrate modified through green in-situ polymerization of bio-derived monomers. This design intentionally preserves the biodegradable backbone of cellulose while introducing solvent resistance through a dynamically crosslinked network that balances solvent stability and degradability.

Performance Beyond Conventional Paper. Traditional paper lacks solvent resistance, exhibits poor wet strength, even disintegrates in polar solvent, and is extremely limited when used as a packaging material. After modification, our material achieves well water contact angle (about 100°) and maintains a tensile strength retention of over 85 MPa after exposure to water for 7 d. Meanwhile, the paper plastic without swelling or dissolve significantly after being immersed in different organic solvents for 7 days (Fig. 4e and Supplementary Fig. 26). These properties surpass conventional paper and approach some thermosets, but crucially the paper plastics combine the recyclable degradability of paper with the water and solvent resistance of thermoset materials.

Revised Fig. 4. Thermal stability, water, and solvent stability of paper plastic. **a**, Coefficient of thermal expansion (CTE) and thermal expansion of polypropylene (PP), polyethylene (PE), polystyrene (PS), poly(lactic acid) (PLA), and paper plastic. **b**, Thermal stability experiment. Comparison of the paper plastic with widely used petroleum-based plastics at 27 $^{\circ}\text{C}$ (R.T.) and 150 $^{\circ}\text{C}$. Compared to 27 $^{\circ}\text{C}$, petroleum-based plastics have already fully softened and deformed at 150 $^{\circ}\text{C}$, while paper plastic still shows no visible change. **c**, Water uptake of paper, NIPU, and paper plastic. (inset) Water contact angle of paper and paper plastic. **d**, Comparison of tensile strength of paper and paper plastics after soaking in water for 7 d. The inset shows the measured tensile strength with error bars representing the \pm SDs ($n = 3$). **e**, Stability of paper plastic in organic solvents after 7 d.

Supplementary Fig. 26 | Analysis of water and solvent resistance. a, Water contact angle of NIPU. b, Change in water contact angle of paper plastic over time. c, Images of the paper plastic after being immersed in different organic solvents for 7 days.

11. - Fig 5d. Once again the comparison of the mechanical performance of the paper plastic with those of commercial plastics, that are thermoplastics is not appropriate. Compare thermoset composites with thermoset composites – with thermosets that are known to interact well with the cellulose (epoxy resin, polyurethane, etc). Compare for instance with a previous work dealing with cellulose-based material/NIPU composites (<https://doi.org/10.1016/j.compositesa.2024.108311>).

Reply to the Reviewer: We thank the Reviewer #3 for raising this important point. We value the suggested reference for its relevance and contribution to our work and cited the literature in the appropriate place in the revised manuscript. Moreover, we also recognize the importance of establishing appropriate comparative frameworks. Therefore, we kindly request the opportunity to elucidate our rationale for the comparison of the mechanical performance.

The motivation for our initial comparison with thermoplastics (e.g., PE, PP) is that our material targets are applied to the packaging sector, which has traditionally been dominated by these polymers. Our paper plastics also exhibit similar or slightly better mechanical strength than these materials. Nevertheless, we fully agree that thermosets provide offer an additional pertinent reference framework. Therefore, we also exhibited the comparative analysis with commercial thermosets (e.g., epoxy, polyurethane) in the revised Fig. 3c. The reviewer's suggestion about the study (<https://doi.org/10.1016/j.compositesa.2024.108311>) have been supplemented in the revised manuscript, the use of materials with high basic mechanical strength and fibers can be used to

composite flax composites reinforced by thermosets with high mechanical strength. This has important heuristic implications for our subsequent research.

Revised Fig. 3. Enhanced mechanical properties and mechanistic insights into the strengthening mechanism of paper plastic. **a**, Stress-strain curves of the cellulose paper and paper plastic. **b**, Comparison of Young's modulus, tensile strength, and toughness of paper (Young's modulus: 1 GPa, strength: 13 MPa, toughness: 0.15 MJ/m^3) and paper plastic (Young's modulus: 3 GPa strength: 126 MPa, toughness: 3.63 MJ/m^3). Values in **b** represent their means \pm SDs from $n = 3$ independent samples. **c**, Ashby diagram of tensile strength vs. Young's modulus for paper plastic compared with typical polymers. **d**, Color-coded two-dimensional maps of stresses carried by cellulose paper at a strain of 5%. **e**, Color-coded two-dimensional maps of stresses carried by paper plastic at a strain of 5%. **f**, SEM image and simulated fracture surface features of the paper. **g**, SEM image and simulated fracture surface features of paper plastic.

However, we have not added a comparison with this material in Fig. 3c, so allow us to illustrate the situation. In the literature (<https://doi.org/10.1016/j.compositesa.2024.108311>), the researchers developed a polyurethane material exhibiting a tensile strength ranging from 80 to 100 MPa, which was subsequently composited with flax fabric to obtain composites. The composites exhibit a highly tensile strength of 438 MPa in the longitudinal direction and 39.8 MPa in the transverse direction. Our paper plastics use isotropic paper (Supplementary Fig. 14) as a matrix, exhibits the same tensile strength (126 MPa) in all directions, and exhibits substantial improvement in mechanical strength compared to the tensile strength of about 1.5 MPa for NIPU polymers alone. The longitudinal direction strength in the literature is significantly better than paper plastics, but our strength in the transverse direction is superior. The strength of the basic materials used in our paper plastics (including NIPU and paper) is lower than that used in the literature. Therefore, we do not think it is appropriate to compare the literature with our work, but we agree that the literature is meaningful to our work and it is also added to the revised manuscript as a key reference. Meanwhile, we would like to emphasize our modification strategy, which can effectively demonstrate significant improvement capabilities for different types of paper (newspapers, printer papers, toilet paper, kraft paper, etc.). In addition, while absolute strength values may appear lower, our system demonstrates exceptional specific performance when normalized to sustainability metrics. The mechanical strength of paper plastics is tenfold that of base paper. Moreover, these materials prepared from biomass, exhibit promising closed-loop recyclability and biodegradability, resulting in a more favorable life cycle assessment.

Updates to the revised manuscript: We thank the Reviewer #3 for raising this important literature. We value the suggested reference for its relevance and contribution to our work. Therefore, the relevant descriptions have been added to the revised manuscript on Page 4 and the literature have been cited in the appropriate place.

“Seychal et al. used epoxy compounds or NIPU to composite with flax to obtain high strength composites. In this work, it is confirmed that the strong hydrogen bond formed between NIPU and the fiber can promote the excellent longitudinally mechanical properties of the flax composite compared with the composite of epoxy resin and flax fiber, which is the result of the interaction between the carbamate bonds in NIPU and the hydroxyl group in the flax fiber. For cellulose paper that bears abundant hydroxy groups on the fibril surface, this dynamic carbamate chemistry is believed to provide vast unexplored opportunities for preparing advanced paper-plastic composites. This also provides a theoretical basis for us to exploit the NIPU modified cellulose paper.”

12. - Fig 5e. Comparison for oxygen barrier properties should be done with the best polymers used for that application, thus ethylene vinyl alcohol copolymers (EVOH) and polyvinyl alcohol (PVOH).

Reply to the Reviewer: We sincerely appreciate the Reviewer's thorough evaluation and valuable feedback. We fully agree with the Reviewer that further optimization of oxygen barrier properties would enhance the material's versatility. As suggested, we have reviewed the literature about ethylene

vinyl alcohol copolymers (EVOH) and polyvinyl alcohol (PVOH) materials¹⁻³, which do exhibit excellent oxygen barrier properties ($1.5 \text{ cm}^3/(\text{m}^2 \cdot 24 \text{ h} \cdot 0.1 \text{ MPa})$ and $6\text{--}38 \text{ cm}^3/(\text{m}^2 \cdot 24 \text{ h} \cdot 0.1 \text{ MPa})$, respectively). Oxygen transmission rate (OTR) is lower than that of paper plastics ($72 \text{ cm}^3/(\text{m}^2 \cdot 24 \text{ h} \cdot 0.1 \text{ MPa})$). However, the primary objective of this work was to address a critical challenge in recyclable materials by developing a novel paper plastic with exceptional mechanical properties, comprehensive stability, and reprocessing ability, which have been identified as key bottlenecks in existing systems. Compared to the commonly used PE/PP ($100\text{--}1000 \text{ cm}^3/(\text{m}^2 \cdot 24 \text{ h} \cdot 0.1 \text{ MPa})$), our paper plastics already meet the application needs. As with the principles that are widely pursued in materials design today, there are often inherent trade-offs between certain properties (e.g., mechanical properties and network dynamics). Therefore, our goal is to develop paper plastics that have advantages in specific aspects, rather than comprehensively surpassing existing materials. But we appreciate the suggestions of Reviewer #3. In our future work, we will try to further improve the oxygen barrier properties of the material to solve this problem.

References:

1. Maes C, Luyten W, Herremans G, Peeters R, Carleer R, Buntinx M. Recent Updates on the Barrier Properties of Ethylene Vinyl Alcohol Copolymer (EVOH): A Review. *Polymer Reviews* 58, 209-246 (2018).
2. Mokwena KK, Tang J. Ethylene Vinyl Alcohol: A Review of Barrier Properties for Packaging Shelf Stable Foods. *Critical Reviews in Food Science and Nutrition* 52, 640-650 (2012).
3. Ge C, Lansing B, Lewis CL. Thermoplastic starch and poly(vinyl alcohol) blends centered barrier film for food packaging applications. *Food Packaging and Shelf Life* 27, 100610 (2021).

Experimental section.

13. - *The mol contents of the different products should be specified, as well as the yields of the products.*

Reply to the Reviewer: We sincerely thank Reviewer #3 for raising this important point. The molar content of the functional groups of each component used in the preparation of paper plastics is supplemented in revised Supplementary Table 1 to facilitate a clearer formulation of the material. In addition, in the preparation of paper plastics, only cyclic carbonate compounds (CSBO and APAC) are prepared, and the rest of the raw materials are commercially available. Meanwhile, in the preparation of cyclic carbonate, the cyclic carbonate compounds we obtained in the high pressure reactor are directly used in the preparation process of subsequent paper plastics without treatment. Most of the product is collected to calculate the yield, except for the compounds remaining in the reactor. Therefore, the yield of the cyclic carbonate compounds can reach 95%.

Updates to the revised manuscript: According to Reviewer #3's comments, we have supplemented the yield of the cyclic carbonate compounds in the revised manuscript on Page 31.

“The cyclic carbonate compounds obtained in the high pressure reactor are directly collected and applied to the subsequent paper plastic preparation process, except for the residue in the reactor chamber, the rest are all products.”

14. - How are recovered the cyclic carbonates after synthesis. Are they degassed to remove dissolved CO₂? Is the catalytic system (TBAI/ascorbic acid) removed, and if yes, how?

Reply to the Reviewer: We thank Reviewer #3 for the constructive comments. At the end of the reaction, the cyclic carbonates are cooled to room temperature and then utilized for the subsequent preparation of non-isocyanate polyurethanes. In addition, the catalytic system used in the preparation of cyclic carbonate compounds has not been removed. Based on our previous experience and the results of Zhang et al.'s work (<https://doi.org/10.1039/d2gc02910c>), these catalysts can have a positive effect on the mechanical properties of the material by promoting the ring-opening reaction during the subsequent curing of the material.

Updates to the revised manuscript: Based on the Reviewer's suggestion, we have added the corresponding description to the revised manuscript on Page 31.

“Paper plastics were obtained directly using unpurified cyclic carbonate. Meanwhile, the catalytic system would decompose to produce tributylamine, which might catalyze the ring opening of the cyclic carbonates and the formation of carbamate bonds.”

15. - Which equipment is used to coat the monomer mixture on the paper, and how is realized the coating?

Reply to the Reviewer: We thank Reviewer #3 for the thoughtful comments. In lab-scale paper plastic preparation, we use a roller brush to apply the monomer mixture to the surface of the paper. The description is supplemented in the preparation section of paper plastics.

Updates to the revised manuscript: According to Reviewer #3's comments, we have added a more careful description of the preparation process of paper plastics in the revised manuscript on Pages 31.

“50 g of APAC, 310 g of CSBO, and 140 g of PEI were first stirred at room temperature for 30 min to obtain a homogeneous mixture. During the mixing of the raw cyclic carbonate compounds with the amine compounds, the temperature of the mixing system is almost stable around room temperature (about 25 °C) within the mixing process. After about 30 minutes of mixing, the viscosity of the mixed system increases to 4000 mPa·s, at which point the mixture is applied to the surface of the paper (500 g, 60 g/m²) with a roller brush and cured through microwave radiation (6 kW, Nanjing Hurui Microwave Technology Development Co., Ltd) for 2 mins to prepare paper plastic.”

16. - *Curing the formulation by microwave radiation-induced curing. How long is the curing step? What is the equipment used and its power? What is the temperature reached by this curing? This latter is important as it will dictate the rate of the reaction but also the content of side products (such as urea bonds).*

Reply to the Reviewer: We thank Reviewer #3 for the thoughtful suggestions. Our curing time of the paper plastic is about two minutes. The curing equipment is the movable tunnel microwave heating and drying equipment with a power of 6 kW produced by Nanjing Hurui Microwave Technology Development Co., Ltd. The operating temperature of the equipment can reach 150 °C, so we can consider the temperature of the curing process to be around 150 °C. This temperature is close to the 160 °C temperature proposed by the reviewer #3 for the formation of side reactions (urea bonds), but due to the short reaction time, the molar ratio of the functional groups of the reaction raw materials is 1:1 (no excess amino residue causes urea bond formation) and FTIR spectra analysis, we believe that the possibility of side reactions during curing is weak.

Updates to the revised manuscript: According to Reviewer #3's comments, we have added a more careful description of the equipment used in the preparation of paper plastics in the revised manuscript on Page 31.

“During the mixing of the raw cyclic carbonate compounds with the amine compounds, the temperature of the mixing system is almost stable around room temperature (about 25 °C) within the mixing process. After about 30 minutes of mixing, the viscosity of the mixed system increases to 4000 mPa·s, at which point the mixture is applied to the surface of the paper (500 g, 60 g/m²) with a roller brush and cured through microwave radiation (6 kW, Nanjing Hurui Microwave Technology Development Co., Ltd) for 2 mins to prepare paper plastic.”

17. - *Synthesis of APAC. I am very surprised by the reaction conditions that are quite harsh, especially for the first step, the reaction of APA with epichlorohydrin. Is there no ring-opening of the epoxide? Due to the complexity of the final cyclic carbonate (APAC), the provided 1H NMR spectrum is not enough to confirm the structure of the molecule. 13C NMR and 2D NMR should be provided to confirm the structure.*

Reply to the Reviewer: We thank the Reviewer #3 for the thoughtful comments. We utilized an excess of epichlorohydrin to serve both as a reaction reagent and solvent in the reaction with APA. During this process, the epoxy group underwent ring-opening while simultaneously forming another epoxy group. We exhibit the mechanism of the reaction of APA with epichlorohydrin. Meanwhile, we also supplemented ¹³C NMR and 2D NMR spectra of APAC to better confirm the molecular structure.

The mechanism of the reaction of APA with epichlorohydrin.

During the reaction between APA and epichlorohydrin, the basic OH⁻ ions attack the epoxy group and open the ring, while the carboxyl compounds lose protons under alkaline conditions to form

carboxylate ions (RCOO^-). The RCOO^- undergoes a substitution reaction with the chlorinated alcohol intermediate, resulting in the formation of ether bonds.

Subsequently, the oxygen of the intermediate hydroxyl group binds to the carbon in the molecular framework, and the removal of the chlorine atom results in the closure of the ring and the formation of a new epoxy group.

¹³C NMR and 2D NMR spectra of APAC.

To better illustrate the structure of the prepared cyclic carbonate compound APAC, we supplemented the ¹³C NMR spectrum and 2D NMR spectrum according to the reviewer's suggestion. As shown in Supplementary Fig. 6, the chemical shifts belonging to the cyclic carbonate groups appear in the NMR spectra, confirming the successful reaction of the epoxide with CO₂ to form a cyclic carbonate.

Supplementary Fig. 6 | NMR spectra of the raw materials (cyclic carbonate APAC). a, ¹H NMR spectrum. b, ¹³C NMR spectrum. c, 2D NMR (COSY) spectrum.

Updates to the revised Manuscript: According to Reviewer #3's comments, we have supplemented the ¹³C NMR spectrum and 2D NMR spectrum in Supplementary Information.

"The chemical shifts belonging to the cyclic carbonate groups appear in the NMR spectra, confirming the successful reaction of the epoxide with CO₂ to form a cyclic carbonate."

18. - For the carbonation, the volume of the reactor should be specified. Are the authors working under constant CO₂ pressure?

Reply to the Reviewer: We thank Reviewer #3 for the constructive comments. The reaction of our epoxy group with CO₂ is carried out in a high-pressure reactor with a chamber volume of 10 L. The reaction process is carried out at a constant pressure of 0.3 MPa, which we show in the "Fabrication of cyclic carbonate" in the manuscript on Page 30.

“CSBO was obtained by adding 100 g ESO, 1.66 g TBAI, and 0.79 g L-ascorbic acid into the pressure reactor, then filling CO₂ to a pressure of the reactor up to 0.3 MPa and reacting at 80 °C for 24 hours.”

19. - Production of the plastic paper. APAC, CSBO and PEI are mixed at room temperature for 30 min before coating the paper. This is quite surprising to mix for a so long period of time when we know that PEI reacts rapidly at room temperature with external cyclic carbonates, thus those of APAC for instance. It is less reactive with the internal cyclic carbonates of CSBO. Is the viscosity of the mixture increased during mixing? Did the authors check the temperature of their mixture over time as the aminolysis of cyclic carbonates is exothermic? It might be dangerous to prepare such mixture on large volumes. Generally the polyamine is mixed rapidly to the cyclic carbonate and then used directly.

Reply to the Reviewer: We thank Reviewer #3 for the constructive comments. The reaction of the external cyclic carbonate groups of APAC with amines is advantageous, but as you mentioned, the internal cyclic carbonate of CSBO is located inside the structure, which to some extent makes it less reactive, so it takes longer to react with amines. In our work, the addition of CSBO was significantly higher than that of APAC, where the external cyclic carbonate group in APAC only accounted for 20% of the total cyclic carbonate group content. Therefore, APAC, CSBO and PEI could be mixed for 30 minutes for the next step of the reaction. In addition, we supplemented the viscosity and temperature changes of CSBO, APAC and PEI during the mixing process.

The viscosity and temperature changes of the mixture during mixing. During the mixing of the raw cyclic carbonate compounds with the amine compounds, the viscosity of the mixture as well as the temperature change were measured by a viscometer. During the mixing process, the temperature of the mixing system is almost stable around room temperature (room temperature is about 25 °C at the time of testing). The viscosity of the system gradually increases with the extension of time, and shows a trend of slow increase in the initial viscosity and rapid increase in the later stage. After about 30 minutes of mixing, the viscosity of the mixed system increases to 4000 mPa·s, at which point the mixture is applied to the surface of the paper and cured to prepare paper plastic.

Fig. R2 The viscosity changes of the mixture during mixing.

Updates to the revised manuscript: According to Reviewer #3's comments, we have added a more careful description of the preparation process of paper plastics in the revised manuscript on Page 31.

“50 g of APAC, 310 g of CSBO, and 140 g of PEI were first stirred at room temperature for 30 min to obtain a homogeneous mixture. During the mixing of the raw cyclic carbonate compounds with the amine compounds, the temperature of the mixing system is almost stable around room temperature (about 25 °C) within the mixing process. After about 30 minutes of mixing, the viscosity of the mixed system increases to 4000 mPa·s, at which point the mixture is applied to the surface of the paper (500 g, 60 g/m²) with a roller brush and cured through microwave radiation (6 kW, Nanjing Hurui Microwave Technology Development Co., Ltd) for 2 mins to prepare paper plastic.”

Supplementary Information

20. Suppl. Fig 8 is not 1H NMR spectra. It represents 13C NMR spectra.

Reply to the Reviewer: We appreciate the issues pointed out by the Reviewer, and we have made changes to the title of Supplementary Fig. 8 (now it is Supplementary Fig. 10) in the revised Supplementary Materials.

Updates to the revised Supplementary Information:

Supplementary Fig. 10. ^{13}C NMR spectra of paper and paper plastic.

21. *Life-cycle assessment (LCA) and technico-economic analysis (TAE). I am not the specialist of these analyses but some of the reported data for the plastic paper seem too good to my opinion. From all LCA/TAE I have read, this is difficult to be better or even close to PE for LCA and TAE of new polymers, especially NIPUs. When I read the protocol for synthesizing APAC from APA, the process is far to be green (large excess of epichlorhydrine, high temperature, etc for the first step), the environmental impact and cost might not be so low. From the reported TAE, paper plastic would be roughly 30% more expensive than PE, this seems very optimistic to me. There is already one interesting report on LCA and TAE of NIPU (<https://doi.org/10.1021/acssuschemeng.4c04046>). Although the cyclic carbonates and polyamines are different, the price of the NIPU is already of more than 5\$/kg. This work should be cited and comparison of LCA/TAE should be done. All these LCA/TAE should be checked by a specialist.*

Reply to the Reviewer: We thank the Reviewer #3 for the thoughtful suggestions. We have reviewed our calculations using the latest available OpenLCA software (2.4.0, released January 2025), the latest available ecoinvent database (ecoinvent v3.11 Cut-Off Unit Processes, released December 2024) and the Environmental Footprint 3.1 methodology. To make the study more detailed, we have extended the system boundary to *cradle-to-grave*, which now includes the manufacturing impacts and those derived from the end-of-life (incineration and fossil carbon emission from the plastic burning, or aerobic composting). Regarding the APAC preparation, our model yields a *climate change* potential of 32.95 kg CO₂ equiv kg⁻¹, which is indeed a considerable footprint. However, it should be noted that its contribution to the overall footprint of the paper plastic is limited because its concentration in the final material is as low as 5.2 wt%. One of the main reasons for the low impact of paper plastic is that it is made from 50% paper, a material with a low carbon footprint. In addition, we have clear end-of-life benefits and the very good mechanical properties of paper-plastic mean that impact is reduced when normalized by tensile strength. We have also included the suggested reference when discussing the LCA results in our manuscript.

Itemized list of response to the reviewers' remarks
(Black: Reviewers' remarks; Blue type: Our response)

Reviewer #1 (Remarks to the Author):

Comments:

[Note from the Editor: this reviewer also looked over the responses to reviewer 3, and felt that these comments were also addressed].

The authors have addressed all of my comments and supplemented the paper with additional experimental results in order to address the concerns of the other reviewers.

Reply to the Reviewer: We thank Reviewer #1 for the suggestion of publication of our work in Nature Communications. We would like to thank all reviewers again for their constructive comments on our work in the previous reviews, which have helped improve the quality of our manuscript.

Reviewer #2 (Remarks to the Author):

Comments:

The authors have adequately addressed all my comments and concerns. I support the publication of this manuscript in its current form, although I recommend ensuring consistency in terminology such as 'Figure' vs. 'Fig.' and units like 'hours' vs. 'h' or 'days' vs. 'd'.

Reply to the Reviewer: We thank Reviewer #2 for the recognition of our work. We would like to thank all reviewers again for their constructive comments on our work in the previous reviews, which have helped improve the quality of our manuscript. We also checked the manuscript to ensure consistency of terminology according to the reviewer's suggestion.